# Efficient PAC Learning of Halfspaces with
# Constant Malicious Noise Rate

**Jie Shen**                                                                JIE.SHEN@STEVENS.EDU
*Stevens Institute of Technology*

**Editors:** Gautam Kamath and Po-Ling Loh

## Abstract

Understanding noise tolerance of machine learning algorithms is a central quest in learning theory. In this work, we study the problem of computationally efficient PAC learning of halfspaces in the presence of malicious noise, where an adversary can corrupt both instances and labels of training samples. The best-known noise tolerance either depends on a target error rate under distributional assumptions or on a margin parameter under large-margin conditions. In this work, we show that when both types of conditions are satisfied, it is possible to achieve *constant* noise tolerance by minimizing a reweighted hinge loss. Our key ingredients include: 1) an efficient algorithm that finds weights to control the gradient deterioration from corrupted samples, and 2) a new analysis on the robustness of the hinge loss equipped with such weights.

**Keywords:** PAC learning, malicious noise

## 1. Introduction

We study the problem of learning halfspaces, a fundamental subject in learning theory (Rosenblatt, 1958; Cortes and Vapnik, 1995). In the absence of noise, it is known that this problem can be efficiently solved via linear programming (Maass and Turán, 1994). However, when some training samples are corrupted, developing efficient algorithms that are resilient to noise becomes challenging. The study of this question has a long history in robust statistics (Huber, 1964) and in learning theory (Valiant, 1985; Angluin and Laird, 1988). In this work, we consider the malicious noise model (Valiant, 1985; Kearns and Li, 1988), possibly the strongest noise, under the probably approximately correct (PAC) learning framework (Valiant, 1984).

Let $\mathcal{X} := \mathbb{R}^d$ be the instance space and $\mathcal{Y} := \{-1, 1\}$ be the label space. The data distribution $D$ is a joint probability distribution over $\mathcal{X} \times \mathcal{Y}$. We denote by $D_X$ the marginal distribution of $D$ on $\mathcal{X}$. The hypothesis class that we aim to learn is the class of homogeneous halfspaces $\mathcal{H} := \{h_w : x \mapsto \text{sign}(w \cdot x), \|w\|_2 = 1\}$. Since any of such halfspace $h_w \in \mathcal{H}$ is uniquely parameterized by a vector $w$, we will often refer to a halfspace $h_w$ by the vector $w$. Given a distribution $D$, the error rate of a halfspace $w$ is defined as $\text{err}_D(w) := \Pr_{(x,y) \sim D}(y \neq \text{sign}(w \cdot x))$.

Under the PAC learning framework, there is a learner and an adversary, and the learning problem under the malicious noise model is described as follows:

**Definition 1 (Learning with malicious noise)** *Let $\epsilon, \delta \in (0, 1)$ be a target error rate and failure probability, respectively. The adversary $\text{EX}(D, w^*, \eta)$ chooses $D$, $w^*$, and $\eta \in [0, \frac{1}{2})$ and fixes them throughout the learning process, such that for all $(x, y) \sim D$, $y = \text{sign}(w^* \cdot x)$. Each time the learner requests a sample from the adversary, with probability $1 - \eta$, the adversary draws a clean sample $(x, y)$ from $D$ and returns it to the learner; with probability $\eta$, the adversary may return an arbitrary dirty sample $(x, y) \in \mathcal{X} \times \mathcal{Y}$. The parameter $\eta$ is referred to as noise rate. The goal of*

*the learner is to find a halfspace $\hat{w} \in \mathcal{H}$, such that with probability at least $1 - \delta$ (over the random draws of samples and all internal randomness of the learning algorithm), it holds that $\mathrm{err}_D(\hat{w}) \leq \epsilon$ for any $D$, $w^*$, and $\eta$.*

It should be noted that the adversary is assumed to have unlimited computational power to search for dirty samples and has full knowledge of the current state and history of the learning algorithm.

A core quest of learning halfspaces with malicious noise is to characterize the noise tolerance of learning algorithms, namely, how large the noise rate $\eta$ can be such that there is still some learning algorithm that PAC learns $\mathcal{H}$. Kearns and Li (1988) showed that the information-theoretic limit of noise tolerance of any algorithm is $\frac{\epsilon}{1+\epsilon}$ if no restriction on $D$ is imposed. They also designed an *inefficient* algorithm with a near-optimal noise tolerance $\eta = \Omega(\epsilon)$, and an efficient algorithm with noise tolerance $\Omega(\epsilon/d)$. A large body of subsequent works then investigated the possibility of achieving optimal noise tolerance with efficient algorithms, usually under various conditions.

Generally speaking, there are two types of conditions that are assumed in order to obtain improved noise tolerance: distributional assumptions and large-margin assumptions. In particular, when the marginal distribution $D_X$ is uniform over the unit sphere, the noise tolerance was improved to $\eta = \widetilde{\Omega}(\epsilon/d^{1/4})$ by Kalai et al. (2005), and this was further improved to $\widetilde{\Omega}(\epsilon^2/\log(d/\epsilon))$ by Klivans et al. (2009). Beyond the uniform distribution, Klivans et al. (2009) derived an algorithm that achieved noise tolerance $\Omega(\epsilon^3/\log^2(d/\epsilon))$ for the family of isotropic log-concave distributions, a broad class that includes Gaussian, exponential, and logistic distributions, among others. However, even under distributional assumptions, the information-theoretic limit $\frac{\epsilon}{1+\epsilon}$ remained unattainable for a long time. In the seminal work of Awasthi et al. (2017), a soft outlier removal scheme was carefully integrated into an active learning framework that gave a near-optimal noise tolerance $\eta = \Omega(\epsilon)$ under isotropic log-concave distributions. Such result was later obtained for a more general noise model, the nasty noise (Bshouty et al., 2002), by Diakonikolas et al. (2018) under the standard Gaussian distribution, which was later relaxed to isotropic log-concave distributions in Shen (2023).

Another line of works assume that the clean samples are separable by the target halfspace with a margin $\gamma > 0$. Servedio (2003) developed a boosting algorithm, termed smooth boosting, that achieved $\eta = \Omega(\epsilon\gamma)$ noise tolerance. Later, Long and Servedio (2011) improved the noise tolerance to $\eta = \Omega(\epsilon\gamma\sqrt{\log 1/\gamma})$ and showed that any algorithm that minimizes a convex surrogate loss can only obtain $\eta = O(\epsilon\gamma)$. One may have observed that when the margin parameter $\gamma$ is small, for example $\gamma = 1/d$, such bounds on noise tolerance are not better than the $\Omega(\epsilon/d)$ given by Kearns and Li (1988). In fact, for the problem of learning large-margin halfspaces, though results established in the literature usually hold for any $\gamma > 0$, it is known that the interesting regime is $\gamma = \Omega(1/\sqrt{d})$ in order to surpass the standard sample complexity bound from VC theory (Shalev-Shwartz and Ben-David, 2014).

Recently, Talwar (2020) showed that the vanilla support vector machines are indeed robust to the malicious noise with noise tolerance $\eta = \Omega(\gamma)$, as far as the $\gamma$-margin condition with $\gamma = \Omega(\frac{\log(1/\epsilon)}{\sqrt{d}})$ and a dense pancake condition are both satisfied, where the latter holds when the underlying distribution $D$ is a mixture of log-concave distributions. This result is significant, in the sense that it was the first time that the large-margin and distributional assumptions are made compatible and are utilized to obtain improved noise tolerance.

We note that in all prior works, the noise tolerance is inherently confined by either the target error rate or the margin. On the other hand, the information-theoretic limit, $\frac{\epsilon}{1+\epsilon}$, does *not* assert the optimality of the established results since it was obtained in a distribution- and margin-free

Table 1: A comparison to state-of-the-art efficient algorithms on learning halfspaces with malicious noise. We obtain the first efficient algorithm with constant noise tolerance.

| Work | Margin | Distribution | Noise tolerance |
|---|---|---|---|
| Long and Servedio (2011) | $\gamma \in (0, 1/7)$ | Not needed | $\Omega(\epsilon\gamma\sqrt{\log(1/\gamma)})$ |
| Awasthi et al. (2017) | Not needed | Isotropic log-concave | $\Omega(\epsilon)$ |
| Theorem 24 of Talwar (2020) | $\gamma = \Omega(\frac{\log(1/\epsilon)}{\sqrt{d}})$ | Log-concave mixture | $\Omega(\gamma)$ |
| **Our work** (Theorem 2) | $\gamma = \Omega(\frac{\log(1/\epsilon)}{\sqrt{d}})$ | Log-concave mixture | $\Omega(1)$ |
| **Our work** (Theorem 5) | Not needed | Separable log-concave mixture | $\Omega(1)$ |

sense. Indeed, the $\Omega(\epsilon\gamma)$ bound already suggested that the nature of the problem may change under structural assumptions on data. This raises a central research question: What is the real noise tolerance of efficient algorithms under standard data assumptions? Since $\eta$ must be less than $\frac{1}{2}$, does there exist an efficient algorithm with $\eta = \Omega(1)$?

## 1.1. Main results

In this work, following Talwar (2020), we assume that both large-margin and distributional conditions are satisfied, and propose an efficient algorithm that achieves noise tolerance $\eta = \Omega(1)$.

**Assumption 1 (Large-margin)** *Any set $S_\mathrm{C}$ of finite clean samples is $\gamma$-margin separable by the target halfspace $w^*$ for some $\gamma > 0$. That is, for all $(x, y) \in S_\mathrm{C}$, $y \cdot w^* \cdot x \geq \gamma$.*

**Assumption 2 (Log-concave mixtures)** *$D_X$ is a uniform mixture of $k$ distributions $D_1, \ldots, D_k$, i.e., $D_X = \frac{1}{k}\sum_{j=1}^k D_j$. In addition, for all $1 \leq j \leq k$, $D_j$ is a log-concave distribution with mean $\mu_j$ and covariance matrix $\Sigma_j$, satisfying $\|\mu_j\|_2 \leq r$ and $\Sigma_j \preceq \frac{1}{d}I_d$ for some $r > 0$.*

**Theorem 2 (Main result)** *There exists an algorithm (Algorithm 1) satisfying the following. For any $\epsilon \in (0, \frac{2}{3}), \delta \in (0, 1)$, if Assumptions 1 and 2 are satisfied with $\gamma \geq \frac{16\log(2/\epsilon)}{\sqrt{d}}$, $r \leq 2\gamma$, $k \leq 64$, and if the malicious noise rate $\eta \leq \frac{1}{2^{32}}$, then by drawing $n = 2^{17} \cdot d \cdot \log^4 \frac{8d}{\epsilon\delta}$ samples from the adversary $\mathrm{EX}(D, w^*, \eta)$, with probability $1 - \delta$, Algorithm 1 returns a halfspace $\hat{w}$ such that $\mathrm{err}_D(\hat{w}) \leq \epsilon$. In addition, Algorithm 1 runs in polynomial time.*

**Remark 3 (Noise tolerance)** *This is the first time that a* constant *noise tolerance is established. As we will see in Section 3, our algorithm simply integrates weights found by a linear program into the hinge loss minimization scheme. Yet, we present novel analysis showing that such paradigm enjoys the strongest robustness. We summarize the comparison to prior works in Table 1.*

**Remark 4 (Condition on $\gamma$)** *We require that $\gamma$ has a logarithmic dependence on $1/\epsilon$ that appears less natural. We believe that this condition can be improved to $\gamma = \Omega(1/\sqrt{d})$ by integrating our algorithm as a subroutine with $\epsilon = \Theta(1)$ into the active learning framework (Awasthi et al., 2017).*

A few more remarks are in order. First, our analysis does not require the distribution $D$ to be centered at the origin. Thus the main result holds also for the class of non-homogeneous halfspaces

$\mathcal{H}' := \{h_{w,b} : x \mapsto \text{sign}\,(w \cdot x + b)\,, w \in \mathbb{R}^d, b \in \mathbb{R}\}$ by embedding instances $x$ as $(x, 1)$; see Talwar (2020) for a more detailed discussion. Second, we believe that it is possible to show the existence of a nearly linear-time algorithm by replacing the hinge loss in Algorithm 1 with a properly chosen smooth loss function; this is left as a future work. Third, it is known that when samples are $\gamma$-margin separable, one may project samples onto an $O(\frac{1}{\gamma^2} \log n)$-dimensional space via random projection (Arriaga and Vempala, 1999; Blum, 2005) while maintaining the margin parameter up to a constant multiplicative factor; this would thus reduce our sample complexity to $O(\frac{1}{\gamma^2} \log^5 \frac{1}{\gamma \epsilon \delta})$.

Our PAC analysis essentially holds when the large-margin condition is empirically satisfied on the set of clean samples, rather than on $D$. Thus, we can alternatively impose a large-margin condition on the centers of the distributions, and show that with high probability, any finite clean sample set satisfies certain large-margin condition. This can also be thought of as an example of how large-margin and distributional assumptions can be made compatible.

**Assumption 3 (Separable log-concave mixtures)** $D_X$ *is a uniform mixture of $k$ distributions* $D_1, \ldots, D_k$, *i.e.* $D_X = \frac{1}{k} \sum_{j=1}^{k} D_j$. *In addition, for all $1 \leq j \leq k$, $D_j$ is a log-concave distribution with mean $\mu_j$ and covariance matrix $\Sigma_j$, satisfying $\left\|\mu_j\right\|_2 \leq r$, $\left|w^* \cdot \mu_j\right| \geq \zeta$, and $\Sigma_j \preceq \frac{1}{d} I_d$ for some $r, \zeta > 0$.*

**Theorem 5** *There exists an algorithm (Algorithm 1) satisfying the following. For any $\epsilon \in (0, \frac{2}{3}), \delta \in (0, 1)$, if Assumption 3 is satisfied with $\zeta \geq \frac{64}{\sqrt{d}} \log^2 \frac{d}{\epsilon \delta}$, $r \in [\zeta, \frac{3}{2}\zeta]$, $k \leq 64$, and if the malicious noise rate $\eta \leq \frac{1}{2^{32}}$, then by drawing $n = 2^{17} \cdot d \cdot \log^4 \frac{8d}{\epsilon \delta}$ samples from the adversary $\text{EX}(D, w^*, \eta)$, with probability $1 - \delta$, Algorithm 1 returns a halfspace $\hat{w}$ such that $\text{err}_D(\hat{w}) \leq \epsilon$. In addition, Algorithm 1 runs in polynomial time.*

Our main results rely on two types of conditions, hence are more stringent than many prior works. Yet, in view of the information-theoretic limit of $\frac{\epsilon}{1+\epsilon}$, it is clear that additional conditions must be assumed in order to obtain the constant rate, though it is largely open to what degree our conditions can be relaxed.

Lastly, we note that little effort was made to optimize the constants in the theorems. In particular, the constant $\frac{1}{2^{32}}$ that upper bounds the noise rate can be slightly improved, but it cannot be made close to the optimal breakdown point of $\frac{1}{2}$ due to inherent limitations of our approach. All omitted proofs can be found in the appendix.

## 1.2. Overview of our techniques

The starting point of our algorithm is the work by Talwar (2020). Though under the malicious noise model, they only obtained $\Omega(\gamma)$ noise tolerance, we observe that a technical barrier that prevents them from getting the $\Omega(1)$ noise tolerance roots in a failure to control the linear sum norm of the sample set, as in their approach, the adversary may construct dirty samples that largely deteriorate such norm. The linear sum norm, which we will describe formally in the next section (see Definition 6), upper bounds the gradient norm of the hinge loss on any sample set. Thus, the main idea of this work is to consider a *reweighted* hinge loss, so that the linear sum norm is reweighted in a similar way. As long as we are able to assign low weight to dirty samples and high weight to clean samples, the linear sum norm would be well controlled, which then implies a PAC guarantee following the framework of Talwar (2020). Thus, our algorithm consists of two primary steps: finding proper weights for all samples, and minimizing a weighted hinge loss.

**1) Finding weights via soft outlier removal.** While finding proper weights by directly minimizing the linear sum norm of dirty samples turns out to be intractable as it requires the knowledge of the identity of dirty samples, we show that the linear sum norm is upper bounded by a reweighted empirical variance among the entire sample set via the Cauchy-Schwarz inequality (Lemma 8). Therefore, we turn to search for weights such that the reweighted empirical variance is properly bounded, which implies that the linear sum norm of the reweighted dirty samples cannot be large (Lemma 13). It is interesting to see that this algorithmic idea indeed coincides with the soft outlier removal scheme proposed by Awasthi et al. (2017) which runs in polynomial time. The difference between our work and theirs lies in how this scheme is utilized: Awasthi et al. (2017) iteratively incorporated it into an active learning framework and used the empirical variance to bound the objective value of hinge loss, while we invoke it only once and use it to bound the gradient norm (more precisely, the linear sum norm).

**2) Analysis on the reweighted hinge loss.** We then analyze the gradient contributions from clean samples $S_C$ and dirty samples $S_D$, respectively, at an optimal solution. Roughly speaking, we show that for any misclassified sample $(x, y)$, the contribution from samples lying at the intersection of $S_C$ and the pancake of $(x, y)$ is upper bounded by the sum of the weights multiplied by $-\gamma$, whereas that from $S_D$ is upper bounded by the linear sum norm. Since the gradient at a global optimal solution is $0$, this shows that for such misclassified sample, the weights of samples in the pancake must be small. Conversely, if the weights are large compared to the linear sum norm, then the sample will be correctly classified. This gives our deterministic result (Theorem 9). Our main result (Theorem 2) follows by showing that after drawing sufficient samples, the soft outlier removal scheme always finds such weights where the linear sum norm is small, and that the weights of samples in the pancake are large provided that the pancake contains many clean samples, which is satisfied under proper distributional assumptions.

### 1.3. Related works

Learning halfspaces in the presence of noise is a fundamental problem in learning theory. Without any assumptions, it is known that establishing PAC guarantees for efficient algorithms is challenging. For example, it is known that even achieving weak PAC guarantees is hard for the agnostic noise (Haussler, 1992; Kearns et al., 1992; Guruswami and Raghavendra, 2006; Feldman et al., 2006; Daniely, 2016). Likewise, little progress was made for the malicious noise if no assumption on the data is imposed (Kearns and Li, 1988; Bshouty, 1998).

By assuming that the marginal distribution $D_X$ is uniform, Gaussian, or isotropic log-concave, a rich set of positive results were established for the malicious noise model. For example, Kalai et al. (2005); Klivans et al. (2009); Awasthi et al. (2017); Diakonikolas et al. (2018) developed efficient algorithms with improved noise tolerance, Shen and Zhang (2021); Shen (2021b) investigated the sample complexity, and Shen (2023) considered fine-grained computational complexity. Notably, this series of works showed that it is possible to PAC learn the class of homogeneous halfspaces in near-linear time with near-optimal sample and label complexity.

The large-margin condition was studied in a rather independent way. A seminal work by Servedio (2003) proposed the smooth boosting algorithm that prevents the algorithm from placing high weight to any sample, an idea that was explored earlier in Domingo and Watanabe (2000). A subsequent work of Long and Servedio (2011) improved the noise tolerance by a logarithmic factor. Interestingly,

a very recent work of Blanc et al. (2024) settled that the known sample complexity bound from smooth boosting is near-optimal.

To the best of our knowledge, Talwar (2020) is the first work that studied the problem of learning halfspaces under both distributional and large-margin conditions. Technically speaking, these two conditions can be made compatible as far as the center of the distribution is far from the decision boundary of the target halfspace and the distribution satisfied certain concentration properties. Talwar (2020) showed that hinge loss minimization is robust against $\Omega(1)$ adversarial label noise, and is also robust against $\Omega(\gamma)$ malicious noise. While the main insight that controlling gradient norm implies robustness shares merit with prior works (Diakonikolas et al., 2021; Prasad et al., 2020), the techniques are inherently different since prior works relied on the smoothness of objective functions to check the spectral norm of certain gradient matrix. Such distinction extends to this work, as we largely follow the framework of Talwar (2020).

Lastly, we note that our research falls into the broad area of algorithmic robustness, which receives a surge of interest in recent years. For example, a large body of works studied the Massart noise (Awasthi et al., 2015; Yan and Zhang, 2017; Diakonikolas et al., 2019; Zhang et al., 2020; Diakonikolas et al., 2020; Chen et al., 2020; Diakonikolas et al., 2022), the adversarial noise (Awasthi et al., 2017; Shen, 2021a), outlier-robust mean estimation (Diakonikolas et al., 2016; Lai et al., 2016; Dong et al., 2019). See references therein and a recent textbook (Diakonikolas and Kane, 2023) for a comprehensive review of the literature.

### 1.4. Roadmap

In Section 2, we collect and define useful notations. Our main algorithms are presented in Section 3, and we provide theoretical analysis in Section 4. We conclude the paper and propose a few open questions in Section 5. The proof details are deferred to the appendix.

## 2. Preliminaries

For two vectors $u$ and $v$, we denote by $u \cdot v$ the standard inner product in the Euclidean space. For a vector $v$, we denote its $\ell_2$-norm by $\|v\|_2$. We write the hinge loss on a sample $(x, y) \in \mathcal{X} \times \mathcal{Y}$ as

$$\ell(w; x, y) := \max \{0, 1 - yw \cdot x\}.$$

Given a sample set $S = \{(x_i, y_i)\}_{i=1}^n$ and a non-negative weight vector $q = (q_1, \ldots, q_n) \in [0, 1]^n$, we define the reweighted hinge loss on $S$ as

$$\ell(w; q \circ S) := \sum_{i=1}^n q_i \cdot \ell(w; x_i, y_i). \tag{2.1}$$

The linear sum norm will play a key role in our analysis. We will need a weighted version of it in our analysis, though for brevity we will still call it linear sum norm.

**Definition 6 (Linear sum norm)** *Given $S = \{(x_i, y_i)\}_{i=1}^n$ and $q = (q_1, \ldots, q_n)$, the linear sum norm of $S$ under the weight vector $q$ is defined as*

$$\mathrm{LinSumNorm}\,(q \circ S) := \sup_{a_1, \ldots, a_n \in [-1, 1]} \left\| \sum_{i=1}^n a_i \cdot q_i \cdot x_i \right\|_2.$$

It is not hard to see that $\mathrm{LinSumNorm}\,(q \circ S)$ serves as an upper bound on the subgradient norm of the reweighted hinge loss $\ell(w; q \circ S)$. For a subset $S' \subset S$, we define $\mathrm{LinSumNorm}\,(q \circ S') := \sup_{i \in S': a_i \in [-1,1]} \left\| \sum_{i \in S'} a_i \cdot q_i \cdot x_i \right\|_2$ by restricting $q$ to the entries corresponding to $S'$. Thus, given an $n$-dimensional vector $q$, the linear sum norm is monotone in $S$: $\mathrm{LinSumNorm}\,(q \circ S') \leq \mathrm{LinSumNorm}\,(q \circ S)$ for any $S' \subset S$. In our analysis, we will mainly be interested in the linear sum norm evaluated on the dirty samples in $S$. To ease our description, we denote the set of clean samples in $S$ by $S_{\mathrm{C}}$, and that of dirty samples by $S_{\mathrm{D}}$, i.e.

$$S = S_{\mathrm{C}} \cup S_{\mathrm{D}}, \quad S_{\mathrm{C}} \cap S_{\mathrm{D}} = \emptyset. \tag{2.2}$$

We remark that the decomposition is for analysis purpose; the learner is unaware of the identity of clean samples.

The dense pancake condition is an important technical tool developed by Talwar (2020) to analyze robustness of the hinge loss. We will also utilize it in this paper.

**Definition 7 (Pancake)** *Let $(x, y) \in \mathcal{X} \times \mathcal{Y}$. Given a unit vector $w \in \mathbb{R}^d$ and a parameter $\tau > 0$, the pancake $P_w^\tau(x, y)$ is defined as*

$$P_w^\tau(x, y) := \left\{ (x', y') \in \mathcal{X} \times \mathcal{Y} : \; \left| y' w \cdot x' - y w \cdot x \right| \leq \tau \right\}.$$

*We say that the pancake $P_w^\tau(x, y)$ is $\rho$-dense with respect to (w.r.t.) a distribution $D$ on $\mathcal{X} \times \mathcal{Y}$ if*

$$\Pr_{(x', y') \sim D} \left( (x', y') \in P_w^\tau(x, y) \right) \geq \rho.$$

*Let $D_1, D_2$ be two distributions on $\mathcal{X} \times \mathcal{Y}$. We say that the pair $(D_1, D_2)$ satisfies the $(\tau, \rho, \beta)$-dense pancake condition if for any unit vector $w \in \mathbb{R}^d$,*

$$\Pr_{(x, y) \sim D_2} \left( P_w^\tau(x, y) \text{ is } \rho\text{-dense w.r.t. } D_1 \right) \geq 1 - \beta.$$

Geometrically speaking, the pancake is a band with width $2\tau$ centered at $(x, y)$ and is orthogonal to $w$. An $\rho$-dense pancake requires that the probability mass of the band is at least $\rho$. The $(\tau, \rho, \beta)$-dense pancake condition says that the pancake of most samples from $D_2$ is $\rho$-dense. In the analysis, $D_2$ will always be the underlying distribution of clean samples, namely the distribution $D$ in Definition 1, and $D_1$ will mostly be the uniform distribution on an empirical set drawn from $D$ (i.e. $S_{\mathrm{C}}$). When it is clear from the context, we will simply say that $(S_{\mathrm{C}}, D)$ satisfies the condition where $S_{\mathrm{C}}$ should be interpreted as the uniform distribution on $S_{\mathrm{C}}$.

## 3. Main Algorithm

We start with a brief review of the approach of Talwar (2020), and identify why their analysis leads to a sub-optimal malicious noise tolerance. We then present our algorithm in detail.

### 3.1. The approach of Talwar (2020)

Talwar (2020) proposed to analyze the robustness of the vanilla hinge loss function, corresponding to the weight vector $q = \mathbf{1} := (1, \ldots, 1)$ in Eq. (2.1). They essentially showed that as far as

$$(1 - \eta)\gamma n \geq \mathrm{LinSumNorm}\,(\mathbf{1} \circ S_{\mathrm{D}}),$$

then one may obtain PAC guarantees along with the large-margin and dense pancake conditions under proper parameter requirements. Nevertheless, by examining the definition of the linear sum norm (Definition 6), one can see that the adversary may construct $S_D$ in such a way – for example, all dirty samples are aligned in a same direction – that the linear sum norm is as large as the size of $S_D$, which is roughly $\eta n$. Therefore, the above condition would read as $\eta \leq O(\gamma)$; this is the noise tolerance announced in their work.

## 3.2. Our algorithm

While such noise tolerance bound seems to be inherent in the analysis of Talwar (2020), based on the above observation, we can see that reweighting samples may diminish the linear sum norm on $S_D$. In the extreme case, had we been able to assign weight $0$ to all samples in $S_D$, the robustness would follow immediately as the linear sum norm becomes $0$. In reality, though, we should not expect that such weight assignment always occurs: the adversary can draw *clean* samples to form $S_D$ and mix them into $S$, in which case any reasonable approach should assign weight $1$ to all samples in $S_D$, just as those in $S_C$. Therefore, the best upper bound that one can expect on the linear sum norm on $S_D$ is the one as if all samples behave very similarly to clean samples (which is roughly $O(|S_D| / \sqrt{d})$). This leads to the following optimization program for finding weights:

$$\text{find } q \in [0,1]^n, \text{ such that } \sup_{a_i : a_i \in [-1,1]} \left\| \sum_{i \in S_D} a_i \cdot q_i \cdot x_i \right\|_2 \text{ is small.} \tag{3.1}$$

The primary trouble is that the above objective function relies on the identity of dirty samples, which is unknown. It turns out that the linear sum norm is monotone in the sense that the norm of a set is always no greater than that of its superset. We thus modify the objective to the linear sum norm on the empirical sample set $S = \{(x_i, y_i)\}_{i=1}^n$, while adding an additional constraint that the sum of $q_i$ is large enough. Suppose that we know an upper bound $\xi$ on the fraction of dirty samples in $S$ (indeed, $\xi = \Theta(\eta)$ by concentration), then we have to assign a weight as large as $1$ to the clean samples, which implies $\sum_{i=1}^n q_i \geq (1 - \xi)n$. We thus obtain

$$\text{find } q \in [0,1]^n, \text{ such that } \sup_{a_i : a_i \in [-1,1]} \left\| \sum_{i \in S} a_i \cdot q_i \cdot x_i \right\|_2 \text{ is small and } \sum_{i=1}^n q_i \geq (1 - \xi)n.$$

The last challenge we have to deal with is the computational efficiency. To our knowledge, it is unclear how to search a feasible $q$ in polynomial time in the above program, since one needs to jointly evaluate over the coefficients $\{a_i\}_{i=1}^n$ and $\{q_i\}_{i=1}^n$. As a technical remedy, we relax this quantity by the Cauchy–Schwarz inequality, showing that the empirical variance serves as an upper bound on the linear sum norm.

**Lemma 8** *Let $S'$ be any set with $m$ samples and $q = (q_1, \ldots, q_m)$ with all $q_i \in [0,1]$. Then* $\text{LinSumNorm}\left(q \circ S'\right) \leq \sqrt{|S'|}\sqrt{\sup_{\|w\|_2 \leq 1} \sum_{i \in S'} q_i \cdot (w \cdot x_i)^2}$.

The intuition underlying the lemma is that the linear sum norm attains its maximum value on $S_D$ when the adversary aligns all dirty samples on a same direction. Such behavior will be signaled by a large empirical variance in that direction, assuming that clean samples are drawn from a well-behaved distribution. Therefore, conversely, if one is able to manage the reweighted empirical variance, then the norm can be controlled.

---

**Algorithm 1** Main Algorithm

---

**Require:** Adversary $\mathrm{EX}(D, w^*, \eta)$, target error rate $\epsilon$, failure probability $\delta$, parameters $\gamma$ and $r$, an upper bound $\eta_0 \geq \eta$.

**Ensure:** A halfspace $\hat{w}$.

1: Draw $n$ samples from $\mathrm{EX}(D, w^*, \eta)$ to form a sample set $\bar{S} = \{(x_i, y_i)\}_{i=1}^n$.

2: Pruning: Remove all samples $(x, y) \in \bar{S}$ with $\|x\|_2 > r + \log \frac{9n}{\delta}$ to form a sample set $S$.

3: Apply Algorithm 2 with inputs $S, \xi \leftarrow 2\eta_0, \bar{\sigma} \leftarrow \sqrt{2(1/d + r^2)}$, and let $q \in [0,1]^n$ be the returned vector.

4: Solve the following reweighted hinge loss minimization program:

$$\hat{v} \leftarrow \underset{\|w\|_2 \leq 1/\gamma}{\arg\min} \ \ell(w; q \circ S).$$

5: **return** $\hat{w} \leftarrow \hat{v} / \|\hat{v}\|_2$.

---

**Algorithm 2** Soft Outlier Removal

---

**Require:** Sample set $S = \{(x_i, y_i)\}_{i=1}^n$, empirical noise rate $\xi$ such that $\xi \geq |S_{\mathrm{D}}| / |S|$, parameter $\bar{\sigma} > 0$.

**Ensure:** A weight vector $q = (q_1, \ldots, q_n)$.

1: Find a vector $q = (q_1, \ldots, q_n)$ that is feasible to the following linear program:

$$\begin{cases} 0 \leq q_i \leq 1, \quad \text{for } i = 1, 2, \ldots, n \\ \sum_{i=1}^n q_i \geq (1 - \xi)n, \\ \sup_{\|w\|_2 \leq 1} \frac{1}{n} \sum_{i=1}^n q_i (w \cdot x_i)^2 \leq \bar{\sigma}^2. \end{cases}$$

2: **return** $q$.

---

We are now in the position to present our algorithm for weight assignment as follows:

$$\text{find } q \in [0,1]^n, \quad \text{such that} \quad \sup_{\|w\|_2 \leq 1} \sum_{i \in S} q_i (w \cdot x_i)^2 \text{ is small and } \sum_{i=1}^n q_i \geq (1 - \xi)n. \tag{3.2}$$

Since the above is a linear program with respect to $q$, it can be solved in polynomial time. It is interesting to note that our algorithmic idea indeed coincides with the soft outlier removal scheme in Awasthi et al. (2017), though the way that we come up with and utilize it is quite different.

Lastly, equipped with the weight vector $q$, we minimize the reweighted hinge loss, as defined in Eq. (2.1). Since we assumed the $\gamma$-margin condition, we need to add the constraint that $\|w\|_2 \leq 1/\gamma$. This results in the following optimization program:

$$\min_{\|w\|_2 \leq 1/\gamma} \ell(w; q \circ S). \tag{3.3}$$

Alternatively, one may also use the constraint $\|w\|_2 \leq 1$, but optimizes a scaled hinge loss function $\max\{0, 1 - \frac{1}{\gamma} y \cdot w \cdot x\}$. The analysis would remain unchanged.

Our full algorithm is shown in Algorithm 1, where the subroutine of soft outlier removal (i.e. weight assignment) is given in Algorithm 2. In Algorithm 1, we perform a pruning step to remove

all samples with large $\ell_2$ norms. This approach is standard (Shen and Zhang, 2021) since with high probability, all clean samples have small $\ell_2$ norm, provided that Assumption 2 is satisfied. We then pass the pruned sample set into the soft outlier removal procedure to assign appropriate weights, ensuring that clean samples receive significantly higher weights compared to dirty samples. Finally, we minimize the reweighted hinge loss to obtain a halfspace.

We recall that in Algorithm 1, the input parameters $\gamma$ and $r$ are defined in Assumptions 1 and 2, respectively. The noise rate upper bound, $\eta_0$, is often available in practical applications; a constant value such as $\eta_0 = \frac{1}{4}$ would work. When invoking Algorithm 2 at the second step of Algorithm 1, the empirical noise rate $\xi$ is set to $2\eta_0$ following the Chernoff bound (see Proposition 22). The $\bar{\sigma}^2$ serves as an upper bound on the empirical variance of clean samples, which translates into an upper bound on the weighted empirical variance of $S$; the value of $\bar{\sigma}^2$ is estimated based on Assumption 2 (see Proposition 14).

### 3.3. Other potential approaches

The way that we control the linear sum norm of $S_{\mathrm{D}}$ is to find proper weights to diminish it, which, as we have shown, can be efficiently solved by applying a soft outlier removal scheme. Such weights will then be utilized to adjust the weight of each individual hinge loss in the objective function. Another approach that we believe is promising is applying the filtering technique (Diakonikolas et al., 2016, 2018; Zeng and Shen, 2023). From a high level, the filtering paradigm compares empirical tail bound to distributional tail bound by projecting samples onto the direction with largest variance, and prunes samples that caused abnormal empirical tail behavior. By repeatedly checking and pruning, it is guaranteed that the empirical variance on all directions is small, which in turn makes the linear sum norm of the remaining samples manageable. It then suffices to minimize the equally weighted hinge loss. We leave detailed investigation to interested readers.

## 4. Performance Guarantees

The analysis of Algorithm 1 is divided into deterministic and statistical parts. We first present a set of results based on deterministic assumptions on the empirical data; these serve as a general recipe for proving the robustness. Then we show that the deterministic conditions are satisfied with high probability under distributional assumptions, leading to Theorem 2.

### 4.1. Deterministic results

Let $S_{\mathrm{C}}$ be the set of clean samples in $S$. Our deterministic analysis of Algorithm 1 relies on the following assumptions.

**Assumption 4 (Feasibility of soft outlier removal)** *The linear program in Algorithm 2 is feasible.*

**Assumption 5 (Dense pancake)** *$(S_{\mathrm{C}}, D)$ satisfies the $(\tau, \rho, \epsilon)$-dense pancake condition.*

**Theorem 9 (Main deterministic result)** *Consider Algorithm 1. Suppose that Assumptions 1 and 4 are satisfied, and Assumption 5 is satisfied for some $\tau \leq \gamma/2$ and*

$$\rho \geq 16 \left( \frac{1}{\gamma \sqrt{d}} + \frac{r}{\gamma} + 1 \right) \sqrt{\eta_0}. \tag{4.1}$$

*Then Algorithm 1 runs in polynomial time and returns $\hat{w}$ such that $\mathrm{err}_D(\hat{w}) \leq \epsilon$.*

The polynomial running time of the algorithm follows from the fact that the hinge loss minimization is a convex program that can be solved in polynomial time (Nesterov, 2004; Boyd and Vandenberghe, 2004) and the following lemma.

**Lemma 10** *Consider Algorithm 2. Suppose that Assumption 4 is satisfied. Then Algorithm 2 returns a feasible solution $q$ in polynomial time.*

The PAC guarantee is established via a set of intermediate results. The following pointwise guarantee is most important.

**Theorem 11** *Consider Algorithm 1. Suppose that Assumption 1 is satisfied. Consider any sample $(x, y) \in \mathcal{X} \times \mathcal{Y}$. If its pancake $P_{\hat{w}}^{\tau}(x, y)$ with $\tau \leq \gamma/2$ is $\rho$-dense with respect to $S_C$ for some $\rho > 4\eta_0$ and if*

$$\frac{1}{4}\gamma \sum_{i \in S_C \cap P_{\hat{w}}^{\tau}(x,y)} q_i > \mathrm{LinSumNorm}\left(q \circ S_D\right), \tag{4.2}$$

*then $(x, y)$ will not be misclassified by $\hat{w}$.*

In Lemma 12, we establish a lower bound on the left-hand side of (4.2) while an upper bound on the right-hand side of (4.2) is established in Lemma 13. By chaining, we obtain a sufficient condition for (4.2), which is exactly the one on $\rho$ in Theorem 9 up to rearrangements and the fact $\xi = 2\eta_0$. Lastly, the pointwise guarantee is extended to the entire support of $D$ via Assumption 5, which leads to Theorem 9.

**Lemma 12** *Consider Algorithm 2. Suppose that Assumption 4 is satisfied. If $(x, y)$ is a sample such that the pancake $P_{\hat{w}}^{\tau}(x, y)$ is $\rho$-dense with respect to $S_C$, then*

$$\sum_{i \in S_C \cap P_{\hat{w}}^{\tau}(x,y)} q_i \geq (\rho - 2\xi)|S|.$$

**Lemma 13** *Consider Algorithm 2. Suppose that Assumption 4 is satisfied. Then we have*

$$\mathrm{LinSumNorm}\left(q \circ S_D\right) \leq \bar{\sigma} \cdot \sqrt{\xi} \cdot |S|.$$

Technically speaking, Lemma 12 provides a guarantee that enough clean samples within the pancake have sufficiently large cumulative weight. To see why this is important, we note that such cumulative weight is almost the magnitude of the gradient norm from the clean samples within the pancake. Lemma 13 shows that if the dirty samples are reweighted by the weight vector $q$ returned by the soft outlier removal, the linear sum norm of the dirty samples can be reduced: recall that the linear sum norm with unweighted samples can be as large as $\xi \cdot |S|$, but now it is $\bar{\sigma} \cdot \sqrt{\xi}$ multiple of $|S|$. Since $\bar{\sigma}$ is roughly the order of $\sqrt{1/d}$ (as we will set $r = \Theta(\gamma) = \Theta(1/\sqrt{d})$), our upper bound in Lemma 13 improves the unweighted version by a factor of $1/\sqrt{d}$. We remark that if $S_D$ consists of independent draws from $D$, then tail bounds for logconcave distributions imply that the unweighted linear sum norm is bounded by $\bar{\sigma} \cdot \xi \cdot |S|$ (Adamczak et al., 2010). Therefore, our result is $\sqrt{\xi}$ factor looser, probably because we relaxed the linear sum norm to the empirical variance for computational efficiency. Nevertheless, such looser scaling seems acceptable in our work, as we are mainly interested in $\xi = \Theta(1)$.

## 4.2. Statistical results

Our statistical analysis aims to show that under Assumption 2, with high probability over the draw of $\bar{S}$ and all internal randomness of the algorithm, Assumptions 4 and 5 are satisfied simultaneously. These combined together with Theorem 9 imply Theorem 2.

**Proposition 14** *Consider Algorithm 1. Suppose that Assumption 2 is satisfied. When $\left|\bar{S}\right| \geq 32d \cdot \log^4 \frac{9d}{\delta}$, with probability at least $1 - \delta$, the linear program in Algorithm 2 is feasible, namely, Assumption 4 is satisfied.*

**Proposition 15** *Suppose that Assumption 2 is satisfied. When $\left|\bar{S}\right| \geq 2048k \cdot d \cdot \log \frac{4d}{\beta\delta}$, then with probability $1 - \delta$, $(S_C, D)$ satisfies the $\left(\frac{8\log(1/\beta)}{\sqrt{d}}, \frac{1}{4k}, 2\beta\right)$-dense pancake condition for any $\beta \in (0, 1/3)$, namely, Assumption 5 is satisfied.*

## 4.3. Proof of Theorem 2

Equipped with both deterministic and statistical results, the main theorem follows immediately from algebraic calculations.

**Proof** Observe that when $\left|\bar{S}\right| \geq 2048k \cdot d \cdot \log^4 \frac{8d}{\beta\delta}$, the following two events hold simultaneously with probability $1 - \delta$: Assumption 4 is satisfied (by Proposition 14); and $(S_C, D)$ satisfies the $\left(\frac{8\log(1/\beta)}{\sqrt{d}}, \frac{1}{4k}, 2\beta\right)$-dense pancake condition (by Proposition 15). From now on, we condition on the two events happening.

By Theorem 9, Algorithm 1 runs in polynomial time and returns $\hat{w}$ with $\text{err}_D(\hat{w}) \leq 2\beta$ if

$$\gamma \geq 2 \cdot \frac{8\log(1/\beta)}{\sqrt{d}}, \quad \frac{1}{4k} \geq 16\left(\frac{1}{\gamma\sqrt{d}} + \frac{r}{\gamma} + 1\right)\sqrt{\eta_0}. \tag{4.3}$$

To get a sufficient condition under which the second inequality above holds, we observe that by $k \leq 64$, we have

$$\frac{1}{4k} \geq \frac{1}{256}.$$

When $\gamma \geq 2 \cdot \frac{8\log(1/\beta)}{\sqrt{d}}$, we have

$$\frac{1}{\gamma\sqrt{d}} + \frac{r}{\gamma} \leq \frac{1}{16\log(1/\beta)} + 2 \leq 4.$$

Thus, Eq. (4.3) holds when $\frac{1}{256} \geq 16 \cdot 4\sqrt{\eta_0}$, i.e., $\eta_0 \leq \frac{1}{2^{32}}$. Lastly, we choose $\beta = \frac{1}{2}\epsilon$ to show that $\text{err}_D(\hat{w}) \leq \epsilon$, and use $k \leq 64$ to obtain the sample size. ∎

## 5. Conclusion and Open Questions

We investigated the problem of learning halfspaces in the presence of malicious noise. We showed that under both large-margin and distributional assumptions, it is possible to design an efficient algorithm with constant noise tolerance, significantly improving upon prior results. It is crucial to investigate if there exist efficient algorithms with noise tolerance arbitrarily close to $\frac{1}{2}$. Another important problem is understanding whether the proposed approach can be adapted to learning sparse halfspaces with sample complexity sublinear in the dimension, and whether linear-time algorithms exist with comparable PAC guarantees.

## Acknowledgments

We thank Kunal Talwar for clarifying technical details of his work. We also thank all reviewers for their constructive comments. This work is supported by NSF-AF-2239376.

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

## Appendix A. Collection of Useful Notations

We use $w^*$ to denote the target halfspace that we aim to learn, with $\|w^*\|_2 = 1$. The optimal solution to the reweighted hinge loss minimization problem (3.3) is denoted as $\hat{v}$, and we use $\hat{w} := \frac{\hat{v}}{\|\hat{v}\|_2}$ to denote the $\ell_2$-normalization of $\hat{v}$. Recall that $\hat{w}$ is the output of our algorithm; see Algorithm 1.

Given a sample $(x, y) \in \mathcal{X} \times \mathcal{Y}$, we will be mostly interested in its pancake at $\hat{w}$ with width $\tau$:

$$P_{\hat{w}}^\tau(x, y) := \left\{ (x', y') \in \mathcal{X} \times \mathcal{Y} : \ \left| y'\hat{w} \cdot x' - y\hat{w} \cdot x \right| \le \tau \right\}.$$

Recall that in Algorithm 1, we denote by $S$ the sample set drawn from $\mathrm{EX}(D, w^*, \eta)$ after pruning. We denote by $S_\mathrm{C}$ the subset of $S$ consisting of clean samples, and by $S_\mathrm{D}$ the remaining samples in $S$. Let

$$S_\mathrm{P}(x, y) := S_\mathrm{C} \cap P_{\hat{w}}^\tau(x, y), \quad S_{\bar{\mathrm{P}}}(x, y) := S_\mathrm{C} \setminus P_{\hat{w}}^\tau(x, y), \tag{A.1}$$

where the dependence on $\hat{w}$ and $\tau$ is omitted since they will be clear from the context. We will often write $i \in S_\mathrm{P}(x, y)$ for $(x_i, y_i) \in S_\mathrm{P}(x, y)$. The loss function $\ell(w; q \circ S)$ can thus be decomposed into three disjoint parts as follows:

$$\ell(w; q \circ S) = \ell(w; q \circ S_\mathrm{P}(x, y)) + \ell(w; q \circ S_{\bar{\mathrm{P}}}(x, y)) + \ell(w; q \circ S_\mathrm{D}), \tag{A.2}$$

where $q \circ T$ should be interpreted as $q_T \circ T$ for $q_T := \{q_i : i \in T\}$ for any $T \subset S$.

## Appendix B. Deterministic Results

### B.1. Analysis of Algorithm 2

Our deterministic analysis relies on Assumption 4, i.e. the linear program in Algorithm 2 is feasible. We will frequently use the parameter setting of $\xi$ such that

$$|S_\mathrm{D}| \le \xi|S|.$$

#### B.1.1. PROOF OF LEMMA 8

**Proof** Observe that

$$\left\| \sum_{i=1}^m a_i \cdot q_i \cdot x_i \right\|_2 = \sup_{\|w\|_2 \le 1} \sum_{i=1}^m a_i \cdot q_i \cdot x_i \cdot w$$

$$\le \sqrt{\sum_{i=1}^m a_i^2} \cdot \sqrt{\sup_{\|w\|_2 \le 1} \sum_{i=1}^m q_i^2 \cdot (w \cdot x_i)^2}$$

$$\le \sqrt{\sup_{\|w\|_2 \le 1} \sum_{i=1}^m q_i^2 \cdot (w \cdot x_i)^2},$$

where the second step follows from the Cauchy-Schwarz inequality, the third step follows from $a_i \in [-1, 1]$ for all $1 \le i \le m$. The lemma follows by taking supremum over all $a_i$'s. ∎

### B.1.2. PROOF OF LEMMA 10

**Proof** Since the linear program in Algorithm 2 is a semi-infinite linear program, it is solvable in polynomial time by the ellipsoid method if there is a polynomial-time separation oracle (Grötschel et al., 2012). The separation oracle was established in Awasthi et al. (2017): for any candidate solution $q$, it first checks whether $0 \le q_i \le 1$ for $i = 1, \dots, n$ and $\sum_{i=1}^{n} q_i \ge (1-\xi)|S|$. Then it checks whether $\sup_{w:\|w\|_2 \le 1} \frac{1}{|S|} \sum_{i \in S} q_i (w \cdot x_i)^2 \le \bar{\sigma}^2$. Note that the supremum can be calculated in polynomial time by singular value decomposition (Golub and Loan, 1996). ∎

### B.1.3. PROOF OF LEMMA 12

The proof follows from the definition of dense pancake as well as the constraints in Algorithm 2.
**Proof** For any $q \in [0,1]^{|S|}$, we have

$$\sum_{i \in S_{\mathrm{D}}} q_i \le |S_{\mathrm{D}}| \le \xi |S| . \tag{B.1}$$

Note that when the pancake is $\rho$-dense with respect to $S_{\mathrm{C}}$, we have $\left|S_{\mathrm{P}}(x,y)\right| \ge \rho |S_{\mathrm{C}}|$ and thus $\left|S_{\bar{\mathrm{P}}}(x,y)\right| \le (1-\rho)|S_{\mathrm{C}}|$. Hence

$$\sum_{i \in S_{\bar{\mathrm{P}}}(x,y)} q_i \le \left|S_{\bar{\mathrm{P}}}(x,y)\right| \le (1-\rho)|S_{\mathrm{C}}| \le (1-\rho)|S| . \tag{B.2}$$

Combining (B.1), (B.2) and the condition $\sum_{i \in S} q_i \ge (1-\xi)|S|$ in Algorithm 2, we get

$$\begin{aligned}
\sum_{i \in S_{\mathrm{P}}(x,y)} q_i &= \sum_{i \in S} q_i - \sum_{i \in S_{\bar{\mathrm{P}}}(x,y)} q_i - \sum_{i \in S_{\mathrm{D}}} q_i \\
&\ge (1-\xi)|S| - (1-\rho)|S| - \xi |S| \\
&\ge (\rho - 2\xi)|S| .
\end{aligned}$$

The proof is complete. ∎

### B.1.4. PROOF OF LEMMA 13

**Proof** Since $q$ is feasible, we have

$$\sum_{i \in S_{\mathrm{D}}} q_i (w \cdot x_i)^2 \le \sum_{i \in S} q_i (w \cdot x_i)^2 \le \bar{\sigma}^2 |S| .$$

Therefore,

$$\sqrt{\sup_{\|w\|_2 \le 1} \sum_{i \in S_{\mathrm{D}}} q_i (w \cdot x_i)^2} \le \bar{\sigma} \sqrt{|S|}.$$

By applying Lemma 8 on $S_{\mathrm{D}}$, we have

$$\sup_{a_i \in [-1,1], i \in S_{\mathrm{D}}} \left\| \sum_{i \in S_{\mathrm{D}}} a_i q_i x_i \right\|_2 \le \sqrt{|S_{\mathrm{D}}|} \cdot \bar{\sigma}\sqrt{|S|} \le \bar{\sigma} \cdot \sqrt{\xi} \cdot |S| .$$

The proof is complete by noting that the left-hand side is the linear sum norm on $S_{\mathrm{D}}$. ∎

## B.2. Analysis of reweighted hinge loss minimization

We denote

$$f(z) := \max\{0, 1 - z\}, \text{ thus } \ell(w; x, y) = f(yw \cdot x). \tag{B.3}$$

We are going to work on hinge loss minimization with a deterministic condition based on first order optimality. Since hinge loss is not differentiable everywhere, we consider the subgradient:

$$\partial f(z) = \begin{cases} \{0\} & \text{if } z > 1, \\ [-1, 0] & \text{if } z = 1, \\ \{-1\} & \text{if } z < 1. \end{cases} \tag{B.4}$$

The subgradient of $\ell(w; x, y)$ equals $\partial_w \ell(w; x, y) = \partial f(yw \cdot x) \cdot yx$.

**Lemma 16** *Consider a sample $(x, y)$ and its pancake $P_{\hat{w}}^\tau(x, y)$ with $\tau \leq \gamma/2$. If $(x, y)$ is misclassified by $\hat{w}$, i.e. $y \cdot \hat{w} \cdot x \leq 0$, then $y_i \hat{v} \cdot x_i < 1$ and thus $\partial f(y_i \hat{v} \cdot x_i) = \{-1\}$ for any $i \in S_P(x, y)$.*

**Proof** For any $i \in S_P(x, y)$, the dense pancake definition indicates that $y_i \hat{w} \cdot x_i \leq y\hat{w} \cdot x + \tau$. Since $y\hat{w} \cdot x \leq 0$ and $\tau \leq \gamma/2 < \gamma$, we have $y_i \hat{w} \cdot x_i < \gamma$. By $\|\hat{v}\|_2 \leq 1/\gamma$ and $\hat{w} = \hat{v}/\|\hat{v}\|_2$, we obtain $y_i \hat{v} \cdot x_i < 1$, which implies $\partial f(y_i \hat{v} \cdot x_i) = \{-1\}$. ∎

**Lemma 17** *For any unit vector $w$ and any $g \in \partial_w \ell(w; q \circ S_D)|_{w=\hat{v}}$, it holds that $g \cdot w \leq \text{LinSumNorm}(q \circ S_D)$.*

**Proof** Observe that

$$\partial_w \ell(w; q \circ S_D)|_{w=\hat{v}} = \sum_{i \in S_D} \partial f(y_i \hat{v} \cdot x_i) y_i (q_i x_i).$$

Since $\partial f(y_i \hat{v} x_i) \subseteq [-1, 0]$, for any $g \in \partial_w \ell(w; q \circ S_D)|_{w=\hat{v}}$ and any unit vector $w$, by the Cauchy-Schwartz inequality,

$$g \cdot w \leq \|g\|_2 \cdot \|w\|_2 \leq \sup_{a_i \in [-1,1], i \in S_D} \left\| \sum_{i \in S_D} a_i y_i (q_i x_i) \right\|_2 = \text{LinSumNorm}(q \circ S_D).$$

The proof is complete. ∎

**Lemma 18** *Consider Algorithm 1. Suppose that Assumption 1 is satisfied. Consider any sample $(x, y) \in \mathcal{X} \times \mathcal{Y}$ and its pancake $P_{\hat{w}}^\tau(x, y)$ with $\tau \leq \gamma/2$. If $(x, y)$ is misclassified by $\hat{w}$, i.e. $y \cdot \hat{w} \cdot x \leq 0$, then for any $g \in \partial_w \ell(w; q \circ S)|_{w=\hat{v}}$, we have*

$$g \cdot w^* \leq -\gamma \sum_{i \in S_P(x, y)} q_i + \text{LinSumNorm}(q \circ S_D).$$

**Proof** We will prove the following three parts individually:

1. For any $g_1 \in \partial_w \ell(w; q \circ S_\mathrm{P}(x,y))|_{w=\hat{v}}$, it holds that $g_1 \cdot w^* \leq -\gamma \sum_{i \in S_\mathrm{P}(x,y)} q_i$.

2. For any $g_2 \in \partial_w \ell(w; q \circ S_{\bar{\mathrm{P}}}(x,y))|_{w=\hat{v}}$, it holds that $g_2 \cdot w^* \leq 0$.

3. For any $g_3 \in \partial_w \ell(w; q \circ S_\mathrm{D})|_{w=\hat{v}}$, it holds that $g_3 \cdot w^* \leq \mathrm{LinSumNorm}\,(q \circ S_\mathrm{D})$.

Note that they together imply the desired result.

To show the first part, consider any $i \in S_\mathrm{P}(x,y)$. Lemma 16 implies $\partial f(y_i \hat{v} \cdot x_i) = \{-1\}$. Therefore,

$$\partial_w \ell(w; q \circ S_\mathrm{P}(x,y))|_{w=\hat{v}} = \sum_{i \in S_\mathrm{P}(x,y)} q_i \cdot \partial f(y_i \hat{v} \cdot x_i) \cdot y_i x_i = - \sum_{i \in S_\mathrm{P}(x,y)} q_i y_i x_i. \tag{B.5}$$

Taking inner product with $w^*$ on both sides and noting that $y_i x_i \cdot w^* \geq \gamma$ for all $i \in S_\mathrm{P}(x,y)$ due to Assumption 1 gives

$$\partial_w \ell(w; q \circ S_\mathrm{P}(x,y))|_{w=\hat{v}} \cdot w^* = - \sum_{i \in S_\mathrm{P}} q_i y_i x_i \cdot w^* \leq -\gamma \sum_{i \in S_\mathrm{P}(x,y)} q_i.$$

We move on to show the second part. Let $i \in S_{\bar{\mathrm{P}}}(x,y) \subset S_\mathrm{C}$. Then we have $y_i w^* \cdot x_i \geq \gamma$ by Assumption 1. Also, we always have $\partial f(y_i \hat{v} x_i) \subseteq [-1, 0]$ in view of the definition of $f$. It thus follows that

$$\partial_w \ell(w; q \circ S_{\bar{\mathrm{P}}}(x,y))|_{w=\hat{v}} \cdot w^* = \sum_{i \in S_{\bar{\mathrm{P}}}} q_i \partial f(y_i \hat{v} \cdot x_i) y_i x_i \cdot w^* \subseteq (-\infty, 0].$$

The last part is an immediate result from Lemma 17. ∎

**Lemma 19** *Consider Algorithm 1 where the solution $\hat{v}$ is such that $\|\hat{v}\|_2 = \frac{1}{\gamma}$. Suppose that Assumption 1 is satisfied. Further assume that $\theta(\hat{w}, w^*) \in (0, \pi/2)$ and let $w' := \frac{w^* - (\hat{w} \cdot w^*)\hat{w}}{\|w^* - (\hat{w} \cdot w^*)\hat{w}\|_2}$. Consider any sample $(x,y) \in \mathcal{X} \times \mathcal{Y}$ and its pancake $P_{\hat{w}}^\tau(x,y)$ with $\tau \leq \gamma/2$. If $(x,y)$ is misclassified by $\hat{w}$, then for any $g \in \partial_w \ell(w; q \circ S)|_{w=\hat{v}}$, we have*

$$g \cdot w' \leq -\frac{1}{2}\gamma \sum_{i \in S_\mathrm{P}(x,y)} q_i + \mathrm{LinSumNorm}\,(q \circ S_\mathrm{D}).$$

**Proof** We will prove the following three parts individually:

1. For any $g_1 \in \partial_w \ell(w; q \circ S_\mathrm{P}(x,y))|_{w=\hat{v}}$, it holds that $g_1 \cdot w' \leq -\frac{1}{2}\gamma \sum_{i \in S_\mathrm{P}(x,y)} q_i$.

2. For any $g_2 \in \partial_w \ell(w; q \circ S_{\bar{\mathrm{P}}}(x,y))|_{w=\hat{v}}$, it holds that $g_2 \cdot w' \leq 0$.

3. For any $g_3 \in \partial_w \ell(w; q \circ S_\mathrm{D})|_{w=\hat{v}}$, it holds that $g_3 \cdot w' \leq \mathrm{LinSumNorm}\,(q \circ S_\mathrm{D})$.

Note that combining them together gives the desired result.

To show the first part, we consider $i \in S_\mathrm{P}(x,y)$. Since $\theta(\hat{w}, w^*) \in (0, \pi/2)$, we have $\alpha := \hat{w} \cdot \hat{w}^* > 0$. By the definition of $w'$, we have

$$y_i w' \cdot x_i = \frac{y_i w^* \cdot x_i - (w^* \cdot \hat{w}) y_i \hat{w} \cdot x_i}{\|w^* - (w^* \cdot \hat{w})\hat{w}\|_2}. \tag{B.6}$$

In addition, for every $i \in S_{\mathrm{P}}(x, y) \subset S_{\mathrm{C}}$, we have that $y_i w^* \cdot x_i \geq \gamma$ by Assumption 1. Also, the pancake definition implies that

$$y_i \hat{w} \cdot x_i \leq y \hat{w} \cdot x + \tau \leq 0 + \gamma/2 = \gamma/2. \tag{B.7}$$

Thus, the numerator of (B.6) satisfies

$$y_i w^* \cdot x_i - (w^* \cdot \hat{w}) y_i \hat{w} \cdot x_i \geq \gamma - \alpha \gamma/2 = (1 - \alpha/2) \gamma.$$

The denominator of (B.6) satisfies

$$\begin{aligned}
\left\| w^* - (w^* \cdot \hat{w}) \hat{w} \right\|_2 &= \sqrt{\left\| w^* - (w^* \cdot \hat{w}) \hat{w} \right\|_2^2} \\
&= \sqrt{\left\| w^* \right\|_2^2 - 2(w^* \cdot \hat{w})^2 + (w^* \cdot \hat{w})^2 \left\| \hat{w} \right\|_2^2} \\
&= \sqrt{1 - 2\alpha^2 + \alpha^2} \\
&= \sqrt{1 - \alpha^2}.
\end{aligned}$$

Thus

$$y_i w' \cdot x_i = \frac{y_i w^* \cdot x_i - (w^* \cdot \hat{w}) y_i \hat{w} \cdot x_i}{\left\| w^* - (w^* \cdot \hat{w}) \hat{w} \right\|_2} \geq \frac{1 - \alpha/2}{\sqrt{1 - \alpha^2}} \gamma \overset{\zeta}{\geq} \frac{\sqrt{3}}{2} \gamma \geq \frac{1}{2} \gamma,$$

where the step $\zeta$ follows from minimizing the expression subject to $\alpha \in (0, 1)$.

On the other hand, Lemma 16 implies $\partial f(y_i \hat{v} \cdot x_i) = \{-1\}$. Thus,

$$\partial_w \ell(w; q \circ S_{\mathrm{P}})|_{w=\hat{v}} \cdot w' = \sum_{i \in S_{\mathrm{P}}(x,y)} q_i \partial f(y_i \hat{v} \cdot x_i) y_i x_i \cdot w' \subseteq \left( -\infty, -\frac{1}{2} \gamma \sum_{i \in S_{\mathrm{P}}(x,y)} q_i \right].$$

Now we move on to show the second part. Consider $i \in S_{\bar{\mathrm{P}}}(x, y) \subset S_{\mathrm{C}}$. If $y_i w' \cdot x_i \geq 0$, then

$$\partial_w \ell(w; q \circ S_{\bar{\mathrm{P}}})|_{w=\hat{v}} \cdot w' = \sum_{i \in S_{\bar{\mathrm{P}}}} q_i \partial f(y_i \hat{v} \cdot x_i) y_i x_i \cdot w' \subseteq (-\infty, 0], \tag{B.8}$$

since $\partial f(\cdot)$ is always non-positive. Now we consider that $y_i w' \cdot x_i < 0$. By the definition of $w'$, we have

$$y_i w' \cdot x_i = \frac{y_i w^* \cdot x_i - (w^* \cdot \hat{w}) y_i \hat{w} \cdot x_i}{\left\| w^* - (w^* \cdot \hat{w}) \hat{w} \right\|_2} < 0,$$

namely,

$$y_i w^* \cdot x_i < (w^* \cdot \hat{w}) y_i \hat{w} \cdot x_i.$$

Since $\theta(\hat{w}, w^*) \in (0, \pi/2)$, we have $\hat{w} \cdot w^* \in (0, 1)$. On the other hand, since $(x_i, y_i) \in S_{\mathrm{C}}$, we have $y_i w^* \cdot x_i > 0$. Plugging into the above inequality gives

$$y_i \hat{w} \cdot x_i > \frac{y_i w^* \cdot x_i}{w^* \cdot \hat{w}} \geq y_i w^* \cdot x_i \geq \gamma$$

where the last inequality follows from Assumption 1. Since $\|\hat{v}\|_2 = 1/\gamma$, we have $y_i \hat{v} \cdot x_i > 1$. It thus follows that $\partial_w \ell(\hat{v}; x_i, y_i) = \{0\}$ as the loss function is the hinge loss. Thus,

$$\partial_w \ell(w; q \circ S_{\bar{P}})|_{w=\hat{v}} \cdot w' = \sum_{i \in S_{\bar{P}}(x,y)} q_i \partial f(y_i \hat{v} \cdot x_i) y_i x_i \cdot w' = \{0\}.$$

Combining the above with Eq. (B.8) proves the second part.

The last part is an immediate result from Lemma 17. ∎

## B.3. Main deterministic results

### B.3.1. PROOF OF THEOREM 11

**Proof** We first note that when $P_{\hat{w}}^{\tau}(x, y)$ is $\rho$-dense with respect to $S_C$ for some $\rho > 4\eta_0$, Lemma 12 showed that

$$\sum_{i \in S_P(x,y)} q_i \geq (\rho - 2\xi)|S| = (\rho - 4\eta_0)|S| > 0, \tag{B.9}$$

where we used the parameter setting $\xi = 2\eta_0$ in Algorithm 1.

Now assume for contradiction that $(x, y)$ is misclassified by $\hat{w}$, i.e. $y\hat{w} \cdot x \leq 0$.

**Case 1.** $\hat{v}$ is in the interior of the constraint set, i.e. $\|\hat{v}\|_2 < 1/\gamma$.

Since $(x, y)$ is misclassified by $\hat{w}$, Lemma 18 tells that

$$g \cdot w^* \leq -\gamma \sum_{i \in S_P(x,y)} q_i + \text{LinSumNorm}\,(q \circ S_D)$$

for any $g \in \partial_w \ell(w; q \circ S)|_{w=\hat{v}}$. On the other hand, by the first-order optimality condition, we have $0 \in \partial_w \ell(w; q \circ S)|_{w=\hat{v}}$. Thus, it must hold that

$$0 \leq -\gamma \sum_{i \in S_P(x,y)} q_i + \text{LinSumNorm}\,(q \circ S_D). \tag{B.10}$$

This in allusion to Eq. (4.2) implies $\sum_{i \in S_P(x,y)} q_i < 0$, but it contradicts Eq. (B.9).

**Case 2.** $\hat{v}$ is on the boundary of the constraint set, i.e. $\|\hat{v}\|_2 = 1/\gamma$.

We first show that $\theta(\hat{w}, w^*) \in (0, \pi/2)$. Observe that $\hat{w} \neq w^*$, as otherwise $(x, y)$ is correctly classified. Hence $\theta(\hat{w}, w^*) \neq 0$. The Karush–Kuhn–Tucker conditions imply that there exists $g \in \partial_w \ell(w; q \circ S)|_{w=\hat{v}}$ such that

$$g + \lambda \hat{w} = 0 \tag{B.11}$$

for some $\lambda \geq 0$. On the other hand, Lemma 18 and Eq. (4.2) together imply $g \cdot w^* < 0$ for all $g \in \partial_w \ell(w; q \circ S)|_{w=\hat{v}}$. Therefore,

$$\lambda \hat{w} \cdot w^* = -g \cdot w^* > 0.$$

This shows $\theta(\hat{w}, w^*) \in (0, \pi/2)$.

Now we are in the position to establish the contradiction. Let $w' := \frac{w^* - (\hat{w} \cdot w^*)\hat{w}}{\left\| w^* - (\hat{w} \cdot w^*)\hat{w} \right\|_2}$ be the component of $w^*$ that is orthogonal to $\hat{w}$ – note that $w'$ is well-defined due to the acute angle between $\hat{w}$ and $w^*$ we just showed. Then it follows that

$$g \cdot w' = -\lambda \hat{w} \cdot w' = 0, \tag{B.12}$$

where the first step is due to Eq. (B.11).

On the other hand, Lemma 19 and the condition (4.2) together imply that

$$g \cdot w' < -\frac{1}{4}\gamma \sum_{i \in S_{\mathrm{P}}(x,y)} q_i < 0, \tag{B.13}$$

where the second inequality follows from Eq. (B.9). This leads to a contradiction to (B.12). ∎

### B.3.2. PROOF OF THEOREM 9

**Proof** Recall that we say $(S_{\mathrm{C}}, D)$ satisfies the $(\tau, \rho, \epsilon)$-dense pancake condition if for any $(x, y)$ drawn from $D$, for any unit $w \in \mathbb{R}^d$, the pancake $P_w^\tau(x, y)$ is $\rho$-dense with respect to $S_{\mathrm{C}}$ with probability $1 - \epsilon$ over the draw of $(x, y)$.

Let $(x, y) \in \mathcal{X} \times \mathcal{Y}$ be such that the pancake $P_{\hat{w}}^\tau(x, y)$ is $\rho$-dense with respect to $S_{\mathrm{C}}$. Our goal is to apply Theorem 11, so we need a sufficient condition for

$$\frac{1}{4}\gamma \sum_{i \in S_{\mathrm{P}}(x,y)} q_i > \mathrm{LinSumNorm}\left(q \circ S_{\mathrm{D}}\right). \tag{B.14}$$

Lemma 12 shows that

$$\sum_{i \in S_{\mathrm{P}}(x,y)} q_i \geq (\rho - 2\xi)|S| = (\rho - 4\eta_0)|S| \geq (\rho - 16\sqrt{\eta_0})|S|, \tag{B.15}$$

where the second step follows from the parameter setting $\xi = 2\eta_0$ and the last step is due to $\eta_0 \leq 1/4$. Lemma 13 tells that

$$\mathrm{LinSumNorm}\left(q \circ S_{\mathrm{D}}\right) \leq \bar{\sigma} \cdot \sqrt{\xi} \cdot |S| \leq 4\left(\frac{1}{\sqrt{d}} + r\right)\sqrt{\eta_0} \cdot |S|. \tag{B.16}$$

Thus, by combining the Eq. (B.15) and Eq. (B.16), we have that Eq. (B.14) holds if

$$\frac{1}{4}\gamma \cdot (\rho - 16\sqrt{\eta_0})|S| \geq 4\left(\frac{1}{\sqrt{d}} + r\right)\sqrt{\eta_0} \cdot |S|, \tag{B.17}$$

which, after rearrangement, is equivalent to

$$\rho \geq 16\left(\frac{1}{\gamma\sqrt{d}} + \frac{r}{\gamma} + 1\right)\sqrt{\eta_0}.$$

Now by Theorem 11, the random sample $(x, y)$ will not be misclassified by $\hat{w}$ with probability $1 - \epsilon$. This is equivalent to

$$\mathrm{err}_D(\hat{w}) \leq \epsilon.$$

For the computational complexity, we note that the reweighted hinge loss is convex and thus can be optimized in polynomial time (Boyd and Vandenberghe, 2004). In addition, Lemma 10 established that Algorithm 2 runs in polynomial time. The proof is complete. ∎

# Appendix C. Statistical Results

We will mainly show that under Assumption 2, both Assumptions 4 and 5 are satisfied with high probability as long as we draw enough samples from $\text{EX}(D, w^*, \eta)$.

To show that Assumption 4 is satisfied (i.e. Proposition 14), we first prove in Proposition 22 that the pruning step in Algorithm 1 will not remove clean samples; and after pruning, the empirical noise rate is upper bounded by $2\eta_0$. We then show by concentration that the empirical covariance of clean samples is at most $\bar{\sigma}^2$, which immediately implies feasibility of the linear program in Algorithm 2.

To show that Assumption 5 is satisfied (i.e. Proposition 15), we simply adapt arguments from Talwar (2020) to our case to complete the proof.

## C.1. Proof of Proposition 14

### C.1.1. ANALYSIS OF PRUNING

**Lemma 20** *Consider Algorithm 1. Suppose that Assumption 2 is satisfied. Then*

$$\Pr_{S' \sim D_X^n} \left( \max_{x \in S'} \|x\|_2 \geq r + \log \frac{3|S'|}{\delta} \right) \leq \delta.$$

**Proof** The proof follows from standard log-concave tail bounds and the union bound.

Consider $D_j$, the $j$-th component of the log-concave mixture. For any $x \sim D_j$, $\sqrt{d}(x - \mu_j)$ is isotropic log-concave. Lemma 28 tells that

$$\Pr_{x \sim D_j} \left( \|\sqrt{d}(x - \mu_j)\|_2 \geq \alpha\sqrt{d} \right) \leq e^{-\alpha+1}, \quad \forall\, \alpha > 0.$$

Let $\alpha = \alpha_0 := \log \frac{3|S'|}{\delta}$. Then with probability at least $1 - \frac{\delta}{|S'|}$, $\|x - \mu_j\|_2 \leq \alpha_0$, implying

$$\|x\|_2 \leq \|x - \mu_j\|_2 + \|\mu_j\|_2 \leq r + \alpha_0.$$

Since $D_X$ is a uniform mixture of $D_1, \ldots, D_k$, we have

$$\Pr_{x \sim D_X} \left( \|x\|_2 \leq r + \alpha_0 \right) = \sum_{j=1}^{k} \frac{1}{k} \cdot \Pr_{x \sim D_j} \left( \|x\|_2 \leq r + \alpha_0 \right) \geq \sum_{j=1}^{k} \frac{1}{k} \cdot \left( 1 - \frac{\delta}{|S'|} \right) = 1 - \frac{\delta}{|S'|}.$$

Taking the union bound over samples in $S'$ completes the proof. ∎

**Lemma 21** *Consider Algorithm 1 with $\eta \leq \eta_0$ for some $\eta_0 \in [\frac{1}{8}, \frac{1}{4}]$. Let $\bar{S}_{\text{C}}$ and $\bar{S}_{\text{D}}$ be the set of clean and dirty samples in $\bar{S}$, respectively. When $|\bar{S}| \geq 32 \log \frac{1}{\delta}$, we have $|\bar{S}_{\text{D}}| \leq 2\eta_0 |\bar{S}|$ and $|\bar{S}_{\text{C}}| \geq (1 - 2\eta_0)|\bar{S}| \geq \frac{1}{2}|\bar{S}|$ with probability $1 - \delta$.*

**Proof** Let $Z_i = \mathbf{1}_{\{x_i \text{ is dirty}\}}$ be the indicator function of the event that $x_i$ is a dirty sample. Since $\Pr(Z_i = 1) = \eta \leq \eta_0$, applying the first inequality in Lemma 26 with $\alpha = 1$ thereof gives

$$\Pr \left( |\bar{S}_{\text{D}}| \geq 2\eta_0 |\bar{S}| \right) \leq \exp \left( -\frac{\eta_0 |\bar{S}|}{3} \right).$$

Since $\eta_0 \geq \frac{1}{8}$, we have $|\bar{S}| \geq 32 \log \frac{1}{\delta} \geq \frac{3}{\eta_0} \log \frac{1}{\delta}$. Thus, with probability $1 - \delta$, we have $|\bar{S}_{\text{D}}| < 2\eta_0 |\bar{S}|$. It then follows that $|\bar{S}_{\text{C}}| \geq (1 - 2\eta_0)|\bar{S}| \geq \frac{1}{2}|\bar{S}|$ where the last step is due to $\eta_0 \leq 1/4$. ∎

**Proposition 22** *Consider Algorithm 1. Suppose that Assumption 2 is satisfied. Let $\bar{S}_{\mathrm{C}}$ and $\bar{S}_{\mathrm{D}}$ be the set of clean and dirty samples in $\bar{S}$, respectively. When $|\bar{S}| \geq 32 \log \frac{2}{\delta}$, with probability $1 - \delta$, the following results hold simultaneously:*

1. *$S_{\mathrm{C}} = \bar{S}_{\mathrm{C}}$, i.e., all clean samples in $\bar{S}$ are intact after pruning.*

2. *$\frac{|S_{\mathrm{D}}|}{|S|} \leq 2\eta_0$, i.e., the empirical noise rate after pruning is upper bounded by $2\eta_0$.*

3. *$|S_{\mathrm{C}}| \geq (1 - 2\eta_0)|\bar{S}| \geq (1 - 2\eta_0)|S|$.*

**Proof** When $|\bar{S}| \geq 32 \log \frac{2}{\delta}$, the following two events hold simultaneously with probability $1 - \delta$: by Lemma 20, for all $x \in \bar{S}_{\mathrm{C}}$, $\|x\|_2 \leq r + \log \frac{6|\bar{S}_{\mathrm{C}}|}{\delta} \leq r + \log \frac{6|\bar{S}|}{\delta}$, and hence $S_{\mathrm{C}} = \bar{S}_{\mathrm{C}}$; by Lemma 21, we have $|\bar{S}_{\mathrm{D}}| \leq 2\eta_0|\bar{S}|$. We condition on these two events.

Since $|S_{\mathrm{D}}| \leq |\bar{S}_{\mathrm{D}}|$, we have

$$\frac{|S_{\mathrm{D}}|}{|S|} = \frac{|S_{\mathrm{D}}|}{|S_{\mathrm{C}}| + |S_{\mathrm{D}}|} \leq \frac{|\bar{S}_{\mathrm{D}}|}{|S_{\mathrm{C}}| + |\bar{S}_{\mathrm{D}}|} = \frac{|\bar{S}_{\mathrm{D}}|}{|\bar{S}_{\mathrm{C}}| + |\bar{S}_{\mathrm{D}}|} = \frac{|\bar{S}_{\mathrm{D}}|}{|\bar{S}|} \leq 2\eta_0.$$

Since $|S_{\mathrm{C}}| = |\bar{S}_{\mathrm{C}}|$ and $|\bar{S}_{\mathrm{D}}| \leq 2\eta_0|\bar{S}|$, we also have

$$|S_{\mathrm{C}}| = |\bar{S}_{\mathrm{C}}| = |\bar{S}| - |\bar{S}_{\mathrm{D}}| \geq (1 - 2\eta_0)|\bar{S}| \geq (1 - 2\eta_0)|S|.$$

The proof is complete. ∎

### C.1.2. PROOF OF PROPOSITION 14

**Proof** The proof largely follows from Shen (2023) which gives near-optimal sample complexity for soft outlier removal. To prove the result, we will show

$$\sup_{\|w\|_2 \leq 1} \frac{1}{|S_{\mathrm{C}}|} \sum_{i \in S_{\mathrm{C}}} (w \cdot x_i)^2 \leq 2\left(\frac{1}{d} + r^2\right). \tag{C.1}$$

Observe that

$$\sup_{\|w\|_2 \leq 1} \frac{1}{|S_{\mathrm{C}}|} \sum_{x \in S_{\mathrm{C}}} (w \cdot x)^2 = \sup_{\|w\|_2 \leq 1} w^\top \left(\frac{1}{|S_{\mathrm{C}}|} \sum_{x \in S_{\mathrm{C}}} xx^\top\right) w = \lambda_{\max}(M),$$

where $M := \frac{1}{|S_{\mathrm{C}}|} \sum_{x \in S_{\mathrm{C}}} xx^\top$ and $\lambda_{\max}(M)$ denotes the maximum eigenvalue. We will apply the matrix Chernoff inequality as stated in Lemma 27 to bound the $\lambda_{\max}(M)$.

In the notation of Lemma 27, we set $\alpha = 1$, $M_i = x_i x_i^\top$, where $x_i$ is the $i$-th instance in the set $S_{\mathrm{C}}$. By Proposition 22, with probability $1 - \delta/3$, all clean samples are retained and $|S_{\mathrm{C}}| \geq (1 - \xi)|S|$. One consequence is that $S_{\mathrm{C}}$ is a set of samples independently drawn from $D_X$. We condition on this event. By Lemma 20, we have

$$\Pr_{S_{\mathrm{C}} \sim D_X^n} \left(\max_{x \in S_{\mathrm{C}}} \|x\|_2 \geq r + \log \frac{9|S_{\mathrm{C}}|}{\delta}\right) \leq \frac{\delta}{3}.$$

Hence with probability $1 - \frac{\delta}{3}$, we have for all $i$ that

$$\lambda_{\max}(M_i) = \|x_i\|_2^2 \leq \left(r + \log \frac{9|S_{\mathrm{C}}|}{\delta}\right)^2 \leq 2r^2 + 2\log^2 \frac{9|S_{\mathrm{C}}|}{\delta}.$$

By Assumption 2, the marginal distribution $D_X$ is a uniform mixture of $k$ log-concave distributions. Then we have

$$
\begin{aligned}
\lambda_{\max}(\mathbb{E}\, M_i) = \left\|\mathbb{E}_{x \sim D_X}\, xx^\top\right\|_2 &= \left\|\frac{1}{k}\sum_{j=1}^k \mathbb{E}_{x \sim D_j}\, xx^\top\right\|_2 \\
&\leq \frac{1}{k}\sum_{j=1}^k \left\|\mathbb{E}_{x \sim D_j}\, xx^\top\right\|_2 \\
&\leq \frac{1}{k}\sum_{j=1}^k \left(\left\|\mathbb{E}_{x \sim D_j}(x - \mu_j)(x - \mu_j)^\top\right\|_2 + \left\|\mu_j \mu_j^\top\right\|_2\right) \\
&\leq \frac{1}{k}\sum_{j=1}^k \left(\frac{1}{d} + r^2\right) \\
&= \frac{1}{d} + r^2.
\end{aligned}
$$

Hence we can set $\mu_{\max} := \lambda_{\max}(\sum_{i \in S_{\mathrm{C}}} \mathbb{E}\, M_i) \leq (\frac{1}{d} + r^2)|S_{\mathrm{C}}|$ in Lemma 27.

Now Lemma 27 asserts that with probability $1 - d \cdot \left(\frac{e}{4}\right)^{\frac{(\frac{1}{d}+r^2)|S_{\mathrm{C}}|}{2r^2 + 2\log^2 \frac{9|S_{\mathrm{C}}|}{\delta}}}$, it holds that

$$\lambda_{\max}(M) \leq 2\left(\frac{1}{d} + r^2\right).$$

It remains to settle the sample size such that the above failure probability equals $\delta/3$. Observe that when

$$\left|\bar{S}\right| \geq 32d \cdot \log^4 \frac{9d}{\delta},$$

we have by Proposition 22

$$|S_{\mathrm{C}}| \geq 16d \cdot \log^4 \frac{9d}{\delta} \geq 16 \cdot \frac{d + r^2 d}{1 + r^2 d} \cdot \log^4 \frac{9d}{\delta}.$$

This implies $d \cdot \left(\frac{e}{4}\right)^{\frac{(\frac{1}{d}+r^2)|S_{\mathrm{C}}|}{2r^2 + 2\log^2 \frac{9|S_{\mathrm{C}}|}{\delta}}} \leq \frac{\delta}{3}$.

Taking the union bound, we obtain that when $\left|\bar{S}\right| \geq 32d \log^4 \frac{9d}{\delta}$, with probability $1 - \delta$, all the following conditions are satisfied simultaneously:

$$|S_{\mathrm{C}}| \geq (1 - \xi)|S|,$$
$$\lambda_{\max}(M) \leq 2\left(\frac{1}{d} + r^2\right).$$

These suggest the existence of a feasible (indeed ideal) solution $q^* = (q_1^*, q_2^*, \ldots, q_{|S|}^*)$ to the linear program in Algorithm 2: $q_i^* = 1$ for every clean sample $(x_i, y_i) \in S_C$ and $q_i^* = 0$ for every dirty sample $(x_i, y_i) \in S_D$. ∎

### C.2. Proof of Proposition 15

The proof in this section largely follows from Talwar (2020), which studied the dense pancake condition over unit ball. We adapt the arguments to log-concave distributions on $\mathbb{R}^d$.

**Lemma 23** *Let $D$ be a distribution over $\mathcal{X} \times \mathcal{Y}$ where the marginal $D_X$ is a log-concave distribution with zero mean and covariance $\Sigma = \frac{1}{d}I_d$. For any $\beta \in (0, 1/3)$, $D$ satisfies the $\left( \frac{4\log(1/\beta)}{\sqrt{d}}, \frac{1}{2}, \beta \right)$-dense pancake condition.*

**Proof** We will first show that over the draw of $(x, y)$, the projection $yw \cdot x$ is well controlled with probability at least $1 - \beta$. Then we show that there are sufficient samples $(x', y')$ that are centered around $(x, y)$ on the direction $w$.

Let $(x, y) \sim D$. Since $D_X$ is a log-concave distribution with zero mean and covariance $\frac{1}{d}I_d$, we know that $\sqrt{d}x$ is an isotropic log-concave random variable. As $y \in \{-1, 1\}$, by the standard tail bound (Lemma 28), we have

$$\Pr_{(x,y)\sim D} \left( \left| yw \cdot \sqrt{d}x \right| \geq \alpha \right) = \Pr_{x\sim D_X} \left( \left| w \cdot \sqrt{d}x \right| \geq \alpha \right) \leq e^{-\alpha+1}$$

for any unit vector $w$. Let $\beta := e^{-\alpha+1}$. Then we have

$$\Pr_{x\sim D_X} \left( |w \cdot x| \leq \frac{\log(1/\beta) + 1}{\sqrt{d}} \right) \geq 1 - \beta.$$

When $0 < \beta < 1/3$, we have $\log(1/\beta) > 1$. Hence

$$\Pr_{x\sim D_X} \left( |w \cdot x| \leq \frac{2\log(1/\beta)}{\sqrt{d}} \right) \geq 1 - \beta.$$

Let $\tau := \frac{4\log(1/\beta)}{\sqrt{d}}$. Then the above inequality tells that with probability $1 - \beta$, we have $|yw \cdot x| \leq \tau/2$. We condition on this event from now on.

On the other hand, using the same steps, we can show that with probability $1 - \beta$ over the draw of $(x', y') \sim D$, we have $\left| y'w \cdot x' \right| \leq \tau/2$. Hence

$$\left| y'w \cdot x' - yw \cdot x \right| \leq \left| y'w \cdot x' \right| + |yw \cdot x| \leq \tau/2 + \tau/2 = \tau.$$

Namely, $P_w^\tau(x, y)$ is $(1 - \beta)$-dense.

Putting together, we have that for $\beta \in (0, 1/3)$, $D$ satisfies the $(\frac{4\log(1/\beta)}{\sqrt{d}}, 1 - \beta, \beta)$-dense pancake condition. Lastly, to ease the expression, we simply note that $1 - \beta \geq 1/2$. ∎

We next show that the same result holds for a log-concave distribution that has non-zero mean and bounded covariance.

**Lemma 24** *Let $D$ be a distribution over $\mathcal{X} \times \mathcal{Y}$ where the marginal $D_X$ is a log-concave distribution with mean $\mu$ and covariance $\Sigma \preceq \frac{1}{d} I_d$. For any $\beta \in (0, 1/3)$, $D$ satisfies the $\left( \frac{4 \log(1/\beta)}{\sqrt{d}}, \frac{1}{2}, \beta \right)$-dense pancake condition.*

**Proof** Since the covariance matrix $\Sigma$ is positive semi-definite, we consider the Cholesky factorization $\Sigma = W^\top W$. Note that the random variable $(X, Y) \sim D$ can be written as $X = W \sqrt{d} X' + \mu$ where $X'$ is isotropic log-concave. By basic linear algebra, $\Sigma \preceq \frac{1}{d} I_d$ is equivalent to $\|\Sigma\|_2 \leq \frac{1}{d}$, and $\|\Sigma\|_2 = \|W^\top W\|_2 = \|W\|_2^2$. Thus we have $\|W\|_2 \leq 1/\sqrt{d}$. Since the function $h(x) := W \sqrt{d} x + \mu$ has Lipschitz constant $\|\sqrt{d} W\|_2 \leq 1$ with respect to the $\ell_2$-norm, by Lemma 23 and Lemma 29, $D$ satisfies the $\left( \frac{4 \log(1/\beta)}{\sqrt{d}}, \frac{1}{2}, \beta \right)$-dense pancake condition. ∎

### C.2.1. PROOF OF PROPOSITION 15

**Proof** By Lemma 24 and Lemma 30, $D$ satisfies the $\left( \frac{4 \log(1/\beta)}{\sqrt{d}}, \frac{1}{2k}, \beta \right)$-dense pancake condition for any $\beta \in (0, 1/3)$. Since $|\bar{S}| \geq 2048 k \cdot d \cdot \log \frac{4d}{\beta\delta} \geq 32 \log \frac{4}{\delta}$, Proposition 22 shows that with probability $1 - \delta/2$,

$$|S_{\mathrm{C}}| \geq \frac{1}{2} |\bar{S}| \geq 24k(d \log d + \log \frac{2}{\beta\delta}) \geq \frac{8k}{1/2} \left( d \log \left( 1 + \frac{2\sqrt{d}}{4 \log 1/\beta} \right) + \log \frac{1}{\beta} + \log \frac{2}{\delta} \right).$$

By Lemma 31, with probability $1 - \delta/2$, $(S_{\mathrm{C}}, D)$ satisfies the $\left( \frac{8 \log(1/\beta)}{\sqrt{d}}, \frac{1}{4k}, 2\beta \right)$-dense pancakes condition for any $\beta \in (0, 1/3)$. Taking the union bound completes the proof. ∎

## Appendix D. Proof of Theorem 5

**Proof** In view of Theorem 2, we need $\gamma \geq 16 \frac{\log(2/\epsilon)}{\sqrt{d}}$ and $|\bar{S}| \geq 2^{17} \cdot d \cdot \log^4 \frac{8d}{\epsilon\delta}$. Lemma 25 shows that we have $\gamma = \frac{3\zeta}{4}$ whenever $\zeta \geq \frac{4}{\sqrt{d}} \log \frac{|\bar{S}|}{\delta}$. Together, it suffices to require

$$\zeta \geq \frac{64}{\sqrt{d}} \log^2 \frac{d}{\epsilon\delta}. \tag{D.1}$$

Theorem 2 also requires $r \leq 2\gamma$. Under $\gamma = \frac{3\zeta}{4}$, this translates into

$$r \leq \frac{3}{2} \gamma. \tag{D.2}$$

Lastly, by the Cauchy-Schwarz inequality, the assumption $|w^* \cdot \mu_j| \geq \zeta$ implies $\|\mu_j\|_2 \geq \zeta$ where we use $\|w^*\|_2 = 1$. Since $r$ is such that $r \geq \|\mu_j\|_2$, we need by chaining

$$r \geq \zeta. \tag{D.3}$$

The above inequality combined with (D.2) gives $r \in [\zeta, \frac{3}{2}\zeta]$. The proof is complete. ∎

**Lemma 25** *Consider Algorithm 1. Suppose that Assumption 3 is satisfied with $\zeta \geq \frac{4}{\sqrt{d}} \log \frac{|\bar{S}|}{\delta}$. Then with probability $1 - \delta$, $\bar{S}_{\mathrm{C}}$ is $\frac{\zeta}{2}$-margin separable by $w^*$, namely, Assumption 1 is satisfied with $\gamma = \frac{3\zeta}{4}$.*

**Proof** Fix an index $j \in \{1, \ldots, k\}$ and consider $x \sim D_j$. Since $\sqrt{d}(x - \mu_j)$ is isotropic log-concave, by Lemma 28, for any $\alpha \geq 0$, we have

$$\Pr_{x \sim D_j} \left( \left| w^* \cdot (x - \mu_j) \right| \geq \alpha/\sqrt{d} \right) = \Pr_{x \sim D_j} \left( \left| w^* \cdot \sqrt{d}(x - \mu_j) \right| \geq \alpha \right) \leq e^{-\alpha+1}.$$

Under the condition $\zeta \geq \frac{4}{\sqrt{d}} \log \frac{|\bar{S}|}{\delta}$, this implies

$$\Pr_{x \sim D_j} \left( \left| w^* \cdot (x - \mu_j) \right| \geq \frac{\zeta}{4} \right) \leq \frac{\delta}{|\bar{S}_{\mathrm{C}}|}.$$

Since $D_X$ is a uniform mixture of $D_1, \ldots, D_k$, we have

$$\Pr_{(x,y) \sim D} \left( \left| w^* \cdot (x - \mu_j) \right| \geq \frac{\zeta}{4} \right) = \sum_{j=1}^{k} \frac{1}{k} \cdot \Pr_{x \sim D_j} \left( \left| w^* \cdot (x - \mu_j) \right| \geq \frac{\zeta}{4} \right)$$

$$\leq \sum_{j=1}^{k} \frac{1}{k} \cdot \frac{\delta}{|\bar{S}_{\mathrm{C}}|}$$

$$= \frac{\delta}{|\bar{S}_{\mathrm{C}}|}.$$

Taking the union bound over $\bar{S}_{\mathrm{C}}$, with probability $1 - \delta$, we have $\left| w^* \cdot (x - \mu_j) \right| \leq \frac{\zeta}{4}$ for all $(x, y) \in \bar{S}_{\mathrm{C}}$. We condition on this event. Since $\left| w^* \cdot \mu_j \right| \geq \zeta$ by Assumption 3, for all $(x, y) \in \bar{S}_{\mathrm{C}}$, we have

$$yw^* \cdot x = \left| w^* \cdot x \right| = \left| w^* \cdot (\mu_j + (x - \mu_j)) \right| \geq \left| w^* \cdot \mu_j \right| - \left| w^* \cdot (x - \mu_j) \right| \geq \zeta - \frac{\zeta}{4} = \frac{3\zeta}{4}.$$

The proof is complete. ■

## Appendix E. Useful Lemmas

**Lemma 26 (Chernoff bounds)** *Let $Z_1, Z_2, \ldots, Z_n$ be $n$ independent random variables that take value in $\{0, 1\}$. Let $Z = \sum_{i=1}^{n} Z_i$. For each $Z_i$, suppose that $\Pr(Z_i = 1) \leq \eta$. Then for any $\alpha \in [0, 1]$*

$$\Pr(Z \geq (1 + \alpha)\eta n) \leq e^{-\frac{\alpha^2 \eta n}{3}}.$$

*When $\Pr(Z_i = 1) \geq \eta$, for any $\alpha \in [0, 1]$*

$$\Pr(Z \leq (1 - \alpha)\eta n) \leq e^{-\frac{\alpha^2 \eta n}{2}}.$$

**Lemma 27 (Matrix Chernoff inequality (Tropp, 2012))** *Let $M_1, M_2, \ldots, M_n$ be $n$ independent, random, self-adjoint matrices with dimension $d$. Suppose that each random matrix $M_i$ satisfies $M_i \succeq 0$ and $\lambda_{\max}(M_i) \leq \Lambda$ almost surely. Let $\mu_{\max} = \lambda_{\max}(\sum_{i=1}^n \mathbb{E}[M_i])$. Then for all $\alpha \geq 0$, with probability at least $1 - d \left[ \frac{e^\alpha}{(1+\alpha)^{1+\alpha}} \right]^{\frac{\mu_{\max}}{\Lambda}}$,*

$$\lambda_{\max}\left( \sum_{i=1}^n M_i \right) \leq (1 + \alpha)\mu_{\max}.$$

**Lemma 28 (Properties of log-concave distributions (Lovász and Vempala, 2007))** *Suppose that $D$ is an isotropic log-concave distribution over $\mathbb{R}^d$. Then we have*

1. *Orthogonal projections of $D$ onto subspaces of $\mathbb{R}^d$ are isotropic log-concave;*

2. *For any given unit vector $w \in \mathbb{R}^d$ and any $\alpha > 0$, $\Pr_{x \sim D}(|w \cdot x| \geq \alpha) \leq e^{-\alpha+1}$;*

3. *For any $\alpha \geq 0$, $\Pr_{x \sim D}(\|x\|_2 \geq \alpha\sqrt{d}) \leq e^{-\alpha+1}$.*

**Lemma 29 (Theorem 17 of Talwar (2020))** *Let $D$ be a distribution on $\mathcal{X} \times \mathcal{Y}$ that satisfies the $(\tau, \rho, \beta)$-dense pancake condition. Suppose that $h : \mathcal{X} \to \mathcal{X}$ is an $L$-Lipschitz function with respect the the $\ell_2$-norm. Then $(h(X), Y)$ is a random variable from a distribution that satisfies the $(L\tau, \rho, \beta)$-dense pancake conditions where $(X, Y) \sim D$.*

**Lemma 30 (Theorem 19 of Talwar (2020))** *Let $D$ be a distribution on $\mathcal{X} \times \mathcal{Y}$. The marginal distribution of $D$ on the instance space $\mathcal{X}$, $D_X$, is a uniform mixture of $k$ distributions $D_1, \ldots, D_k$, i.e. $D_X = \frac{1}{k}\sum_{j=1}^k D_j$. If for all $1 \leq j \leq k$, $D_j$ satisfies $(\tau, \rho, \beta)$-dense pancake condition, then $D$ satisfies the $(\tau, \rho/k, \beta)$-dense pancake condition.*

**Lemma 31 (Theorem 20 of Talwar (2020))** *Let $D$ be a distribution that satisfies the $(\tau, \rho, \beta)$-dense pancake condition. Let $S_{\mathrm{C}}$ be a set samples drawn from $D$. If*

$$|S_{\mathrm{C}}| \geq \frac{8}{\rho}\left( d \log\left(1 + \frac{2}{\tau}\right) + \log\frac{1}{\beta} + \log\frac{1}{\delta}\right),$$

*then with probability $1 - \delta$, $(S_{\mathrm{C}}, D)$ satisfies $(2\tau, \rho/2, 2\beta)$-dense pancake condition.*

