# OpenReview forum: "Efficient PAC Learning of Halfspaces with Constant Malicious Noise Rate"
_algorithmiclearningtheory.org/ALT/2025/Conference — ALT 2025_

### Official Review · Reviewer_thhX · 2024-11-08
**Constant noise tolerance halfspace learner under strong assumptions**

**Rating:** 6
**Confidence:** 4

**Review:**

Contributions: The paper designs an algorithm for learning (realizable) halfspaces up to 0-1 error epsilon, under the assumption that an eta fraction of samples are maliciously corrupted (i.e., the adversary is randomly assigned corruptible samples rather than being able to inspect the samples first). The paper is the first to give a polynomial-time algorithm for this problem in the regime eta = Omega(1). To do so, it uses several (fairly strong) distributional assumptions, including: a margin gamma a little larger than d^{-1/2}, and the distribution being a mixture of O(1) components all of which have small means and sub-isotropic covariances.

The paper essentially modifies an argument due to [Talwar, '20] which achieves all of the conditions above except requires eta = O(gamma). The observation is that all outliers can be in the same direction, inducing a gradient change of roughly eta * n, and the ERM algorithm is only tolerant of size-gamma * n perturbations. Instead, the paper uses a classic idea from robust statistics, observing that the gradient norm is boundable by the covariance operator norm. They therefore run a filtering technique by solving an SDP (an idea which has appeared before in e.g., the robust estimation literature, see "High-Dimensional Robust Mean Estimation in Nearly-Linear Time" for the first example to my knowledge, and "Sever: A Robust Meta-Algorithm for Stochastic Optimization" for the idea of filtering to control gradient sizes). Their final algorithm simply combines the SDP filtering step with an ERM step.

I think the paper is reasonable. The conceptual message is nice (there is a highly-structured setting where PAC learning under Omega(1) malicious noise is tractable), but the techniques are a bit on the straightforward side. Much of the analysis in the paper is either directly adapted from [Talwar, '20], or based on empirical concentration ideas standard in robust statistics. I think the paper would be more interesting if the paper was considering a setting beyond the restricted assumptions in [Talwar, '20] or gave a faster algorithm.

Towards that end, I would have liked to see more discussion about if the algorithm can be implemented efficiently. It seems like it should be doable -- hinge loss ERM (Step 4, Algo 1) is a well-studied problem in convex optimization, so it would be good to discuss what off-the-shelf runtimes would give (should achieve nd / poly(gamma)?) Regarding Algo 2, I think that this problem is a box-constrained packing SDP and hence should also be solvable using off-the-shelf tools from the literature in nearly-linear time. For example, see the reductions in the aforementioned "High-Dimensional Robust Mean Estimation in Nearly-Linear Time" paper, Section 4, or the specific box-constrained packing SDP solver in "Robust Sub-Gaussian Principal Component Analysis and Width-Independent Schatten Packing", Proposition 2.

A few smaller comments.
1. Is "instances" standard terminology for x in R^d? I've seen "features" more.
2. It'd be great to discuss a bit the relationship of malicious noise to other noise models for the unfamiliar reader. (Is it the same thing as Massart noise, except rather than only control the labels, the adversary also gets to control the instances of a random eta fraction?)
3. Assumption 2 seems rather strong... it extends to only a constant number of components with very small means, which feels almost like a single near-isotropic component with large margin. Could the author include a bit of discussion of what is technically more challenging about extending to this setting rather than the single-component setting?
4. The statement of Theorem 2 has some weird asymmetry. I'm not sure why the constants are specified everywhere except the random c.
5. "as far as" -> "as long as" in Section 3.1?
6. The authors refer to (4) as a "linear program" and a "semi-infinite linear program". Isn't it just a (packing) semidefinite program?
7. I think by far the most important open question is to remove distributional assumptions from the current work, or show they're necessary for the results. The assumptions used in the work (margin, essentially isotropic) are quite strong and unlikely to hold in realistic instances. This isn't really mentioned at all in Section 5, which makes me a little concerned with the paper's messaging, given its main upside is conceptual (extending to an important parameter regime) rather than technical in my viewpoint.

**Paper Award:**

No

---

### Official Review · Reviewer_8Gww · 2024-11-09
**Learning halfspaces with both margin and distribution assumptions**

**Rating:** 8
**Confidence:** 4

**Review:**

This work studies learning halfspaces in the presence of malicious noise, where an adversary can corrupt both instances and labels of training samples.

$\textbf{Summary of relevant literature.}$ The work provides very detailed information about prior works, which are important to interpret the contribution of this work. The literature is summarized below.

Under the setup, the goal is often to characterize the maximum amount of noise (potentially as a function of relevant parameters such as accuracy parameter $\epsilon$ and dimension $d$) one can tolerate so that the error of the learning algorithm remains bounded from above by $\epsilon$.

Without any further assumption, the theoretically optimal answer to the above is $\epsilon / (1 + \epsilon) = \Theta(\epsilon)$.

A long line of work has focused on designing efficient algorithms under distributional assumptions.  In particular, the bound $\Omega(\epsilon)$ is shown to be attainable by polynomial time algorithms under isotropic log-concave distributions.

Another line of work instead imposes margin assumptions on the clean sample points, and obtains algorithms that could tolerate at most $\tilde \Omega(\epsilon \gamma)$ fraction of noise, where $\gamma$ is the margin size. Somewhat surprisingly, this bound has already surpassed the assumption-free info-theoretic bound of $\epsilon / (1 + \epsilon)$ when $\gamma$ is sufficiently large, showing that the margin assumption has fundamentally changed the nature of the problem information theoretically.

$\textbf{Main Result.}$ Building on top of the prior work, this work shows that, under both the margin and a pancake-style distribution assumption (which is satisfied by mixture of log-concave distributions), the noise-tolerance rate can be improved all the way to $\Omega(1)$, a function independent of any other relevant parameters of the problem.

$\textbf{Technique.}$ The technique builds on top of the prior work of Talwar (2020), which analyzes hinge loss minimization under the same setup. In particular, the author observes that the main bottleneck of the prior work is that the adversary may construct samples that result in large linear sum norm — a quantity closely related to the subgradient norm of the hinge loss function, and hence also the estimation error.  Motivated by this, their algorithm uses the idea of soft outlier removal, developed in the context of designing a polynomial time algorithm with optimal error tolerance under distribution assumptions, to find weights to control the empirical variance of the sample sets. This turns out to be sufficient to control the linear sum norm of the sample, and they just need to perform weighted hinge loss minimization to conclude the algorithm.

$\textbf{Strengths.}$ The paper is very well-written. The algorithm proposed is natural and effective, but leads to very surprising results. Overall, I think this is a good contribution to the community.

$\textbf{Weakness}.$ The fact that the margin needs to scale poly-logartihmically with the accuracy parameter is a bit unsatisfying, and the authors leave it as future work whether such dependency can be improved.

Question: It seems like the result is for homogeneous halfspace. But since the distribution assumption does not require the covariance to be of full-rank, does that mean one can also handle general halfspaces by simplifying embedding the bias as an extra coordinate?

**Paper Award:**

No

---

### Official Review · Reviewer_dBzf · 2024-11-10
**Efficient PAC Learning of Halfspaces with Constant Malicious Noise Rate**

**Rating:** 6
**Confidence:** 3

**Review:**

This paper studies the PAC learning of halfspaces under a constant malicious noise rate. Specifically, for any sample (x,y) drawn from a given distribution, an adversary can replace it with an arbitrary pair with some probability. This paper presents an algorithm that requires 1/sqrt{d} margin and allows any constant noise level. The core idea of the algorithm is to use a linear program to reweight the queries, thereby minimizing the impact of corrupted samples. This approach builds on the framework established by Talwar 2020 with some new ingredients.

Overall, this is a nice result. However, the contribution appears to be incremental, and the technical advancements are also not particularly interesting, as they largely build on previous work.

**Paper Award:**

No

---

### Author Rebuttal · Authors · 2024-11-19

**Response to Reviewer dBzf:**

We thank the reviewer for the nice summary. It seems, however, that the reviewer overlooked our main contribution. Breaking the information-theoretic noise limit of O(eps) under natural assumptions is one of the most important open questions in the literature. Not only do we break it, but also we obtain best possible noise tolerance (up to constant factor).

Regarding technical advancement, it is surprising that such a simple mechanism already gives the best known result. It is foreseeable that this work will inspire a rich line of follow-up study to, for example, 1) weaken the assumptions; 2) improve sample and label complexity; 3) extend to other robust learning problems.

**Response to Reviewer 8Gww:**

We thank the reviewer for the very detailed summary of our contribution and encouraging feedback.

Regarding the polylog(1/eps) factor in the condition of the margin, we are working on new algorithms including active learning and boosting to remove it - the key observation is that under these frameworks, the algorithm of this paper can serve as a subroutine with eps being a small constant.

Regarding learning of non-homogeneous halfspaces, the reviewer is correct that we can embed the bias as an extra coordinate. We will add a remark to discuss such extension.

---

### Author Rebuttal · Authors · 2024-11-19

**Response to Reviewer thhX (1/2):**

We thank the reviewer for the constructive comments. A fine-grained analysis of running time was also a question that we thought about when preparing the manuscript, and we are more than happy to discuss it. First of all, our outlier removal approach (Alg 2) can be implemented in nearly linear time via online regret minimization; see Dong et al 2019 for original ideas in mean estimation and Shen 2023 for extension to learning halfspaces. As a side comment, the running time of the SDP approach of Cheng et al 2019 incurs a high scaling in $1/\epsilon$, and had thus been superseded. The analysis of hinge loss minimization is more subtle than it appears to be. If the optimal solution lies on the boundary of the constraint set, it is straightforward to apply projected stochastic gradient descent (SGD) to get linear-time computational cost. Yet, if it lies within, then our analysis requires finding a solution with sufficiently small subgradient (the analysis in our submission assumes zero subgradient solution, but it is not hard to tailor it to allow small deviation). Now the trouble is that since hinge loss is non-smooth, a numerical solution with low objective value may not enjoy low subgradient. One potential remedy is to consider smooth approximations to the hinge loss and apply projected SGD.

Regarding our technical contribution, while *in hindsight*, our algorithm appears natural, we note that Talwar 2020 was published many years ago yet no progress was made towards this important problem. Also, we remark that our idea to control the gradient size is quite different from prior works: they essentially require the loss function to be smooth in order to evaluate the spectral norm of a gradient matrix, which quickly breaks down in hinge loss. Our approach is to upper bound the gradient size by empirical variance of instances. Lastly, we consider the simplicity of our algorithm as advantage, which will inspire a rich line of follow-up study to, for example, 1) weaken the assumptions; 2) improve sample and label complexity; 3) extend to other robust learning problems.

---

### Author Rebuttal · Authors · 2024-11-19

**Response to Reviewer thhX (2/2):**

For minor comments:

- The use of ''Instance'' dates back to as early as Kearns and Li 1988.
- Malicious noise behaves more like the adversarial label noise, and is thus much stronger than the Massart noise. We will discuss various noise models in more detail in the revision.
- In Assumption 2, we chose to work with mixture models since they are more general than single component. The requirement on the mean is not that strong though: the mean indeed matches the standard deviation (both are at the order of $1/\sqrt{d}$), so that instances mostly fall into unit ball - note that this is needed to define margin. We can alternatively assume the standard deviation being 1, under which the mean needs to be a constant; our analysis still follows.
- $c$ is introduced mainly to indicate $r = O(\gamma)$. Yes, we can pick a concrete value such as $c = 1$.
- Regarding necessity of our assumptions: Kearns and Li constructed a data distribution $D$ to show the information-theoretic noise tolerance is $O(\epsilon)$. Thus, to obtain $\Omega(1)$ noise tolerance, we must assume extra conditions. Since our work is the first attempt, it is largely open how much we can weaken the current assumptions. On the other hand, we note that both large margin and logconcavity are broadly assumed in the literature (see page 2). We will add such discussion in our revision.

---

### Meta-Review · Area_Chair_eazf · 2024-12-15

**Recommendation:** Accept
**Confidence:** 5

**Metareview:**

The submission studies the problem of efficient PAC learning under malicious noise, where an adversary can corrupt instances and labels of training examples.
It gives an algorithm that minimizes a reweighted hinge loss and achieves constant noise tolerance under both a margin assumption and a distributional assumption (satisfied by mixtures of log-concave distributions). The paper builds on the prior work of Talwar (NeurIPS 2020), which analyzed hinge loss minimization under a setup, but the submission is the first to achieve a noise tolerance that is an absolute constant.
The paper is a good fit for the ALT program.
I encourage the author(s) to incorporate the comments of the reviewers in the final version.

**Paper Award:**

No